# Thalamic spindles and Up states coordinate cortical and hippocampal co-ripples in humans

Charles W. Dickey[1,2,3]*, Ilya A. Verzhbinsky[1,2], Sophie Kajfez[4], Burke Q. Rosen[1,5], Christopher E. Gonzalez[1], Patrick Y. Chauvel[6,7,8], Sydney S. Cash[9], Sandipan Pati[10‡], Eric Halgren[3,11‡]*

1 Neurosciences Graduate Program, University of California San Diego, La Jolla, California, United States of America, 2 Medical Scientist Training Program, University of California San Diego, La Jolla, California, United States of America, 3 Department of Psychiatry and Behavioral Sciences, Stanford University, Palo Alto, California, United States of America, 4 Department of Radiology, University of California San Diego, La Jolla, California, United States of America, 5 Department of Neuroscience, Washington University in St. Louis, St. Louis, Missouri, United States of America, 6 Aix-Marseille Université, Marseille, France, 7 INSERM, Institut de Neurosciences des Systèmes UMR 1106, Marseille, France, 8 APHM (Assistance Publique–Hôpitaux de Marseille), Timone Hospital, Marseille, France, 9 Department of Neurology, Massachusetts General Hospital, Boston, Massachusetts, United States of America, 10 Department of Neurology, University of Texas Health Science Center at Houston, Houston, Texas, United States of America, 11 Department of Neurosciences, University of California San Diego, La Jolla, California, United States of America

‡ These authors are joint senior authors on this work.
* cdickey@health.ucsd.edu (CWD); ehalgren@health.ucsd.edu (EH)

## Abstract

In the neocortex, ~90 Hz ripples couple to ~12 Hz sleep spindles on the ~1 Hz Down-to-Up state transition during non-rapid eye movement sleep. This conjunction of sleep waves is critical for the consolidation of memories into long-term storage. The widespread co-occurrences of ripples ("co-ripples") may integrate information across the neocortex and hippocampus to facilitate consolidation. While the thalamus synchronizes spindles and Up states in the cortex for memory, it is not known whether it may also organize co-ripples. Using human intracranial recordings during NREM sleep, we investigated whether cortico-cortical co-ripples and hippocampo-cortical co-ripples are either: (1) driven by directly projected thalamic ripples; or (2) coordinated by propagating thalamic spindles or Up states. We found ripples in the anterior and posterior thalamus, with similar characteristics as hippocampal and cortical ripples, including having a center frequency of ~90 Hz and coupling to local spindles on the Down-to-Up state transition. However, thalamic ripples rarely co-occur or phase-lock with cortical or hippocampal ripples. By contrast, spindles and Up states that propagate from the thalamus strongly coordinate co-ripples in the cortex and hippocampus. Thus, thalamo-cortical spindles and Up states, rather than thalamic ripples, may provide input facilitating spatially distributed co-rippling that integrates information for memory consolidation during sleep in humans.

**Data Availability Statement:** The source data are provided with the manuscript. The custom code that supports the findings of this study is available

in the following repository: https://zenodo.org/records/13909725.

**Funding:** This work was supported by the National Institute of Mental Health (NIMH) through awards 1RF1MH117155-01 (EH) and T32 MH020002 (EH), and by the Office of Naval Research Multidisciplinary University Research Initiatives Program (ONR-MURI) through award N00014-16-1-2829 (EH). The funders had no role in the study design, data collection and analysis, decision to publish, or preparation of the manuscript.

## Introduction

Ripples are brief high-frequency oscillations of the local field potential (LFP). In rodent hippocampus during non-rapid eye movement (NREM) sleep, ripples mark the replay of spatiotemporal firing patterns that are crucial in the consolidation of declarative memories [1–4]. In humans, approximately 90 Hz ripples are found in the cortex during both NREM and wakefulness [5,6]. Cortical ripples couple to cortical and hippocampal ripples preceding recall [6,7] and entrain the replay of neuron firing sequences established during encoding [8,9]. Co-rippling increases between the ventral wordform area and language areas during reading, and strongly increases between language areas prior to correct semantic judgments, supporting a more general role in cortical integration [10].

Spindles are 10 to 16 Hz oscillations that occur during NREM and are important for memory consolidation [11]. They are generated through interactions of intrinsic currents and local circuits in the thalamus [12,13] and are projected to all cortical areas [14,15], often on the Down-to-Up state transition [16], timing that is important for memory consolidation [17]. Down states, where cortical neurons are silent, and Up states, where cortical neurons fire at near-waking levels, comprise the slow oscillation and K-complex of human NREM [18,19]. Recurrent thalamo-cortical projections support the initiation, synchronization, and termination of spindles and Down-to-Up states [15,20,21]. Critical to a possible role in consolidation, cortical ripples occur during spindles just prior to the Up state peak [5,22] and may mark the replay of cell-firing encoding patterns learned in prior waking [9]. Furthermore, hippocampal ripples are coordinated with cortical ripples [6–8], spindles, Down states, and Up states [23–25].

Crucially, human cortical ripples co-occur and phase-lock across long distances in multiple lobes in both hemispheres [6]. Neither co-occurrence nor phase-locking decreases with separation between rippling locations, implying either a network of densely connected coupled cortical oscillators, or a central subcortical structure that broadly triggers and synchronizes cortical ripples. The thalamus is a prime candidate for this role given its direct anatomical connections with both the hippocampus and cortex [26], as well as its ability to modulate and synchronize spindles and Down-to-Up states across widespread cortical areas [15,27]. Consolidation during NREM in mice is enhanced by optogenetic stimulation of the thalamus that evokes cortical spindles co-occurring with cortical Up states and hippocampal sharpwave ripples [28]. Thus, the thalamus could synchronize ripples, either directly by generating and widely projecting ripples to the cortex, or indirectly by projecting spindles or Up states, which in turn evoke ripples in multiple cortical and hippocampal locations.

Ripples have not been previously studied in the human thalamus. We analyzed a rare collection of intracranial recordings from non-epileptogenic and non-lesioned anterior and posterior thalamus, with simultaneous recordings in the hippocampus and cortex obtained from patients with epilepsy undergoing seizure focus localization. We asked if physiological ripples occur in the human thalamus, and if so, what are their characteristics and relationships with cortical and hippocampal ripples. Specifically, we tested if thalamic ripples co-occur and phase-lock with cortical or hippocampal ripples, as would be expected if thalamic ripples directly synchronize ripples in these regions. We found approximately 90 Hz ripples in the human thalamus with similar density (rate of occurrence) and duration as cortical and hippocampal ripples, and like cortical ripples, thalamic ripples couple to local spindles and occur on the Down-to-Up state transition during NREM. However, thalamic ripples are only weakly related to cortical and hippocampal ripples, whereas thalamo-cortical spindles and Up states are associated with increased probability of co-rippling between cortical sites. Thus, the thalamus does not appear to project ripples to the cortex, but instead the spindles and Up states that it projects to the cortex act to synchronize widespread ripple co-occurrences. In other words,

the thalamus appears to synchronize the overall occurrence of cortical co-ripples but not their individual waves.

## Results

### Ripples are found in the human thalamus during NREM sleep

We detected ripples in the thalamus, cortex, and hippocampus of 13 patients undergoing seizure focus localization with intracranial electrodes (S1 and S2 Tables and S1 and S2 Figs). All channels included in the study were trans-gray matter bipolar derivations to minimize effects due to volume conduction, and localized to non-lesioned, non-epileptogenic tissue. Electrodes were implanted in anterior thalamic nuclei, projecting mainly to lateral, medial and orbital prefrontal cortices, and the cingulate gyrus (S1 Table). Electrodes were also implanted in posterior thalamic nuclei, projecting mainly to posterior cortical areas. Sleep staging was performed to select NREM. Channels and epochs were only included if they did not contain frequent interictal spikes or artifacts. Ripples were detected using a previously published method [5,6], requiring increased amplitude and at least 3 distinct oscillation cycles within 70 to 100 Hz. Putative ripples were rejected if they contained possible interictal spikes or artifacts.

We found that in addition to the cortex and hippocampus, ripples were identified in the human anterior and posterior thalamus during NREM (Fig 1A–1D). These ripples were centered focally around ~90 Hz with a preceding increase in delta (0.1 to 4 Hz) and concurrent increase in spindle (10 to 16 Hz) band power. Thalamic ripple detections did not appear to be due to volume conduction since adjacent non-thalamic bipolar channels in the white matter did not demonstrate ripples at the times of the thalamic ripples (S3 Fig). Furthermore, ripples were very infrequently detected in adjacent non-thalamic bipolar channels in the white matter (S4 Fig).

### Thalamic ripples have similar characteristics as hippocampal and cortical ripples

Anterior thalamic ripples during NREM had a mean and standard deviation density (occurrence rate) of 19.8 ± 4.6 min$^{-1}$, peak 70 to 100 Hz analytic amplitude of 1.79 ± 0.91 µV, oscillation frequency of 92.4 ± 0.6 Hz, and duration of 62.6 ± 5.9 ms (Fig 1E–1H and S3 Table). Posterior thalamic ripples had a density of 11.3 ± 2.2 min$^{-1}$, amplitude of 1.95 ± 0.34 µV, frequency of 90.2 ± 0.4 Hz, and duration of 93.3 ± 23.0 ms. Cortical ripples had an average density of 21.9 ± 2.5 min$^{-1}$, amplitude of 4.17 ± 1.69 µV, frequency of 90.1 ± 0.5 Hz, and duration of 65.3 ± 3.6 ms (S5 Fig shows cortical ripples from patients with posterior thalamic leads). Hippocampal ripples had an average density of 21.3 ± 4.7 min$^{-1}$, amplitude of 11.39 ± 7.23 µV, frequency of 88.5 ± 1.3 Hz, and duration of 74.8 ± 7.9 ms. Overall, these characteristics were consistent when analyzed across individual ripples (S6 Fig) and across channels for individual patients (S7 Fig). While thalamic ripples had smaller amplitudes than cortical or hippocampal ripples, they were easily distinguished from the baseline signal (Fig 1Aiii and 1Ciii), which was also smaller. These characteristics were similar to those of hippocampal and cortical ripples, with the exception of hippocampal ripples having larger amplitudes (Fig 1E–1H and S3 Table). Posterior thalamic ripples had a slightly lower density, similar amplitudes, slightly lower frequencies, and longer durations than anterior thalamic ripples. The relative sizes of ripples we detected in thalamus, cortex, and hippocampus are consistent with the degrees of lamination of neurons and synaptic inputs in the 3 structures, as well as previous observations that human ripples are larger in hippocampus than cortex [5], and that spindles and Down states are larger in cortex than thalamus [15].

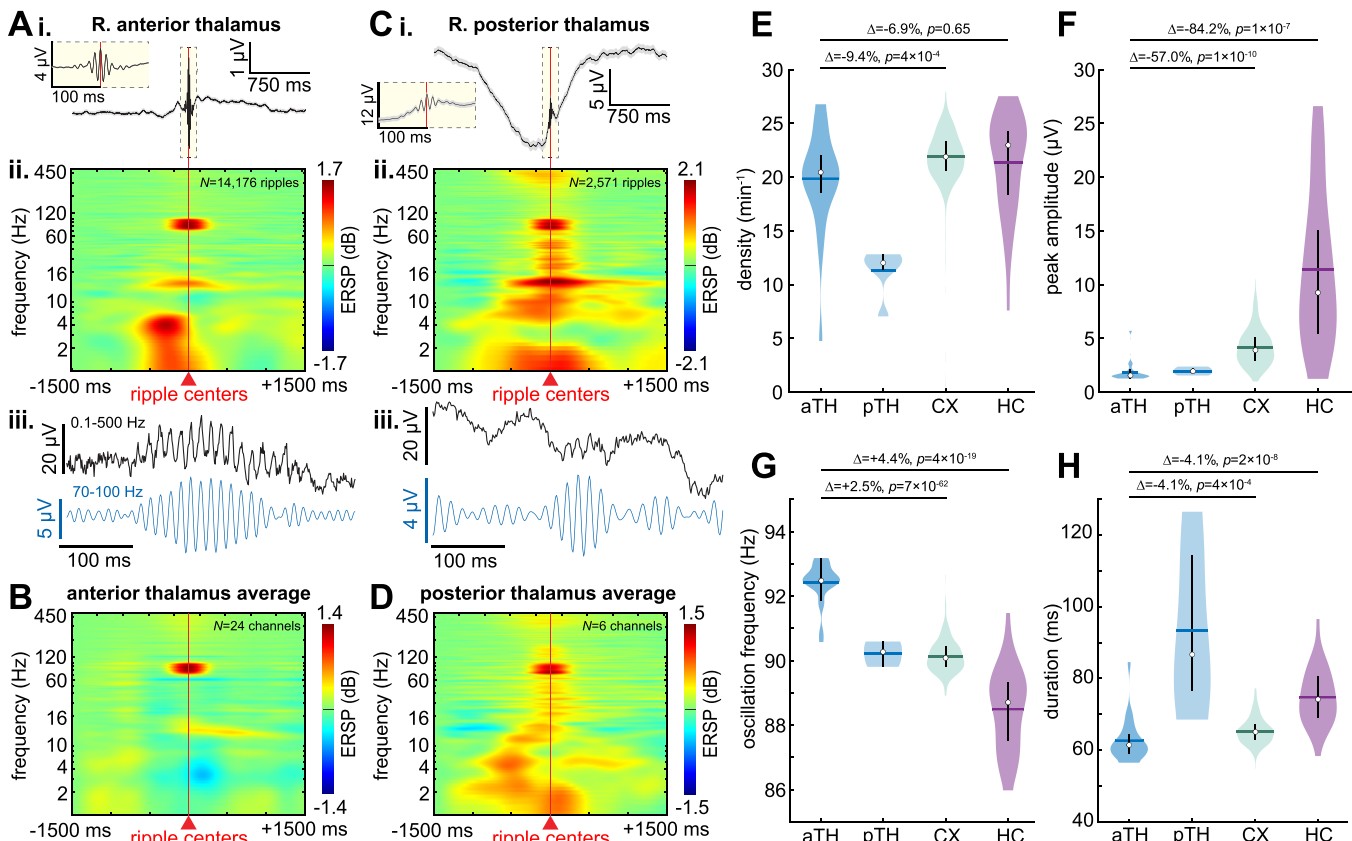

**Fig 1. Human thalamic ripples during NREM have similar characteristics as cortical and hippocampal ripples.** (**A**) Ripples in one anterior thalamic bipolar channel: (**i**) average and SEM broadband LFP, (**ii**) average time-frequency, and (**iii**) example single sweep broadband (black) and 70–100 Hz bandpass (blue). In the time-frequency plot, note the focally increased power at ~90 Hz, concurrent with increased ~12 Hz spindle band power, shortly following increased ~4 Hz delta power. (**B**) Grand channel average time-frequency of all anterior thalamic channels ($N$ = 24 channels from patients 1–10). (**C, D**) Same as A and B except posterior thalamic ripples ($N$ = 6 channels from patients 11–13). (**E–H**) Anterior thalamic, posterior thalamic, cortical, and hippocampal ripple densities (**E**), peak 70–100 Hz analytic amplitudes (**F**), oscillation frequencies (**G**), and durations (**H**) during NREM across all channels ($N_{aTH}$ = 24, $N_{pTH}$ = 6, $N_{CX}$ = 261, $N_{HC}$ = 31 channels). Note the consistent oscillation frequencies of ~90 Hz. Horizontal lines, means; circles, medians; vertical lines, interquartile ranges. Statistics performed with linear mixed-effects models with patient as random effect. $P$-values FDR-corrected for multiple comparisons across all channels from all patients. Distributions across individual ripples are shown in S6 Fig. aTH, anterior thalamus; CX, cortex; FDR, false discovery rate; HC, hippocampus; LFP, local field potential; NREM, non-rapid eye movement sleep; pTH, posterior thalamus; SEM, standard error of the mean. Source data are available in S1 Data.

## Thalamic ripples couple to local spindles on the Down-to-Up state transition

It is thought that the Down state is initiated in the cortex and then projects to the thalamus, where hyperpolarization releases h and T currents leading to the generation of a thalamic spindle that is then projected back to the cortex at the time of the Down-to-Up state transition [15]. This precisely timed sequence is important for memory consolidation. We found that, on average, anterior and posterior thalamic ripples occur approximately 225 to 275 ms following thalamic Down state peaks (Fig 2A–2D; $N$ = 9/24 anterior thalamic channel pairs with significant modulations, post-FDR $p < 0.05$, randomization test; significance across all channel pairs within −500 to 0 ms: $p = 2 \times 10^{-19}$, $z = 8.9$, and 0 to 500 ms: $p = 0.47$, $z = -6 \times 10^{-2}$, one-sided Wilcoxon signed-rank test; $N$ = 2/6 posterior thalamic, −500 to 0 ms: $p = 6 \times 10^{-5}$, $z = 3.9$, and 0 to 500 ms: $p = 0.99$, $z = 2.2$). Of the 9 anterior thalamic channels with significant modulations, 7 had a significant order preference, all with Down states preceding ripples (post-FDR

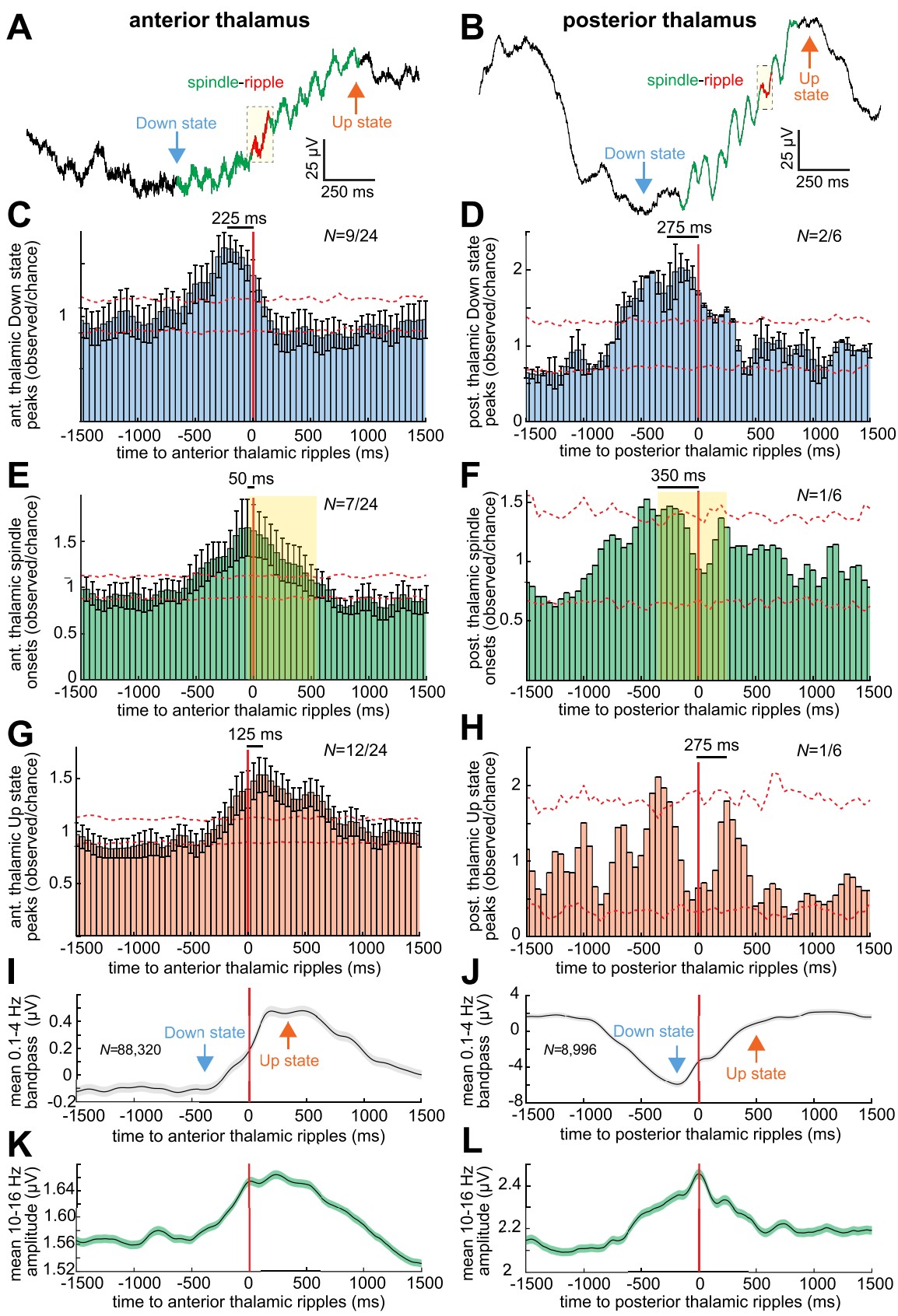

**Fig 2. Thalamic ripples occur on the local Down-to-Up state transition during spindles, similar to cortical ripples. (A, B)** Example broadband single sweep anterior (**A**) and posterior (**B**) thalamic ripple (in red) occurring during a spindle (in green) on the Down-to-Up state transition. (**C, D**) Average and SEM times of anterior (**C**) and posterior (**D**) thalamic Down state peaks relative to thalamic ripples at $t = 0$ across channels that showed significant coupling, sub-selected to demonstrate effect time courses ($N_{aTH} = 9/24$ and $N_{pTH} = 2/6$ significant channels, post-FDR $p < 0.05$, randomization test with shuffled controls). Dashed error shows 98% confidence interval of the null distribution. See S8 Fig for results across all channels. (**E, F**) Same as C and D except thalamic spindle onsets ($N_{aTH} = 7/24$ and $N_{pTH} = 1/6$). Shaded box denotes the average spindle interval. (**G, H**) Same as C and D except thalamic Up state peaks ($N_{aTH} = 12/24$, $N_{pTH} = 1/6$). (**I, J**) Average and SEM anterior (**I**) and posterior (**J**) thalamic ripple-locked 0.1–4 Hz delta bandpass ($N_{aTH} = 88,320$ and $N_{pTH} = 8,996$ ripples). (**K, L**) Same as I and J except 10–16 Hz spindle band analytic amplitude. Source data are available in S2 Data.

$p < 0.05$, binomial test, expected value = 0.5). Similarly, of the 2 posterior thalamic channels with significant modulations, both had a significant order preference with Down states preceding ripples. Anterior thalamic ripples tended to occur during thalamic spindles, shortly following their onsets (Fig 2E; $N = 7/24$, −500 to 0 ms: $p = 1 \times 10^{-16}$, $z = 8.2$, and 0 to 500 ms: $p = 2 \times 10^{-12}$, $z = 6.9$). Posterior thalamic ripples also coupled to thalamic spindles, albeit with a greater delay (approximately 350 ms) following spindle onset (Fig 2F; $N = 1/6$, −500 to 0 ms: $p = 3 \times 10^{-4}$, $z = 3.4$, and 0 to 500 ms: $p = 0.92$, $z = -1.4$). Anterior and posterior thalamic ripples occurred approximately 125 ms and approximately 275 ms, respectively, preceding thalamic Up state peaks (Fig 2G and 2H; anterior thalamus $N = 12/24$, −500 to 0 ms: $p = 0.97$, $z = 1.92$, and 0 to 500 ms: $p = 9 \times 10^{-17}$, $z = 8.24$; posterior thalamus $N = 1/6$, −500 to 0 ms: $p = 0.028$, $z = -1.9$, and 0 to 500 ms: $p = 0.09$, $z = 1.3$). Of the 12 anterior thalamic channels with significant modulations, 9 had a significant order preference, all with thalamic ripples preceding thalamic Up state peaks. Results from all channels confirm these relationships (S8 Fig). Some thalamic channels also had ripples that occurred during cortical spindles following the cortical Down state and preceding the cortical Up state (S9 Fig). Overall, these data show that thalamic ripples tend to occur following the local Down state, with anterior thalamic ripples occurring preceding the local Up state (Fig 2I and 2J), especially during spindles (Fig 2K and 2L).

## Thalamic ripples infrequently and weakly couple to and co-occur with cortical and hippocampal ripples

Previously, we showed that ripples frequently and strongly co-occur between cortical sites (i.e., cortico-cortical co-ripples) in widespread networks, and also (less strongly) between the hippocampus and cortex (i.e., hippocampo-cortical co-ripples) [6]. One possible mechanism underlying this is a thalamo-cortical driving circuit. Therefore, we tested if thalamic ripples couple (within ±250 ms) or co-occur ("co-ripple"; ≥25 ms overlap) with cortical or hippocampal ripples. Confirming our previous results in a different dataset, we found frequent and strong cortico-cortical ripple coupling (Fig 3A and 3B). By contrast, we found that anterior and posterior thalamic ripples infrequently and weakly coupled with cortical ripples (Fig 3C and 3D). Replicating our previous findings, hippocampo-cortical ripple coupling was frequent but weak (Fig 3E). By contrast, we found infrequent and weak coupling between anterior thalamic and hippocampal ripples (Fig 3F). Thalamic ripple auto-correlation did not reveal rhythmic entrainment by Down states or Up states (S10 Fig). Results from all channel pairs confirm these relationships (S11 Fig). Notably, there was a significantly increased probability of cortico-cortical co-rippling given rippling on a different randomly selected cortical channel compared to the probability of rippling on a randomly selected anterior thalamic channel (mean ± SEM: 0.18 ± 0.003 versus 0.09 ± 0.001%; $p = 1 \times 10^{-233}$, $t(3897) = 34.8$; one-sided paired $t$ test) as well as posterior thalamic rippling on a randomly selected channel (mean ± SEM: 0.67 ± 0.09 versus 0.17 ± 0.03%; $p = 6 \times 10^{-7}$, $t(27) = 6.2$).

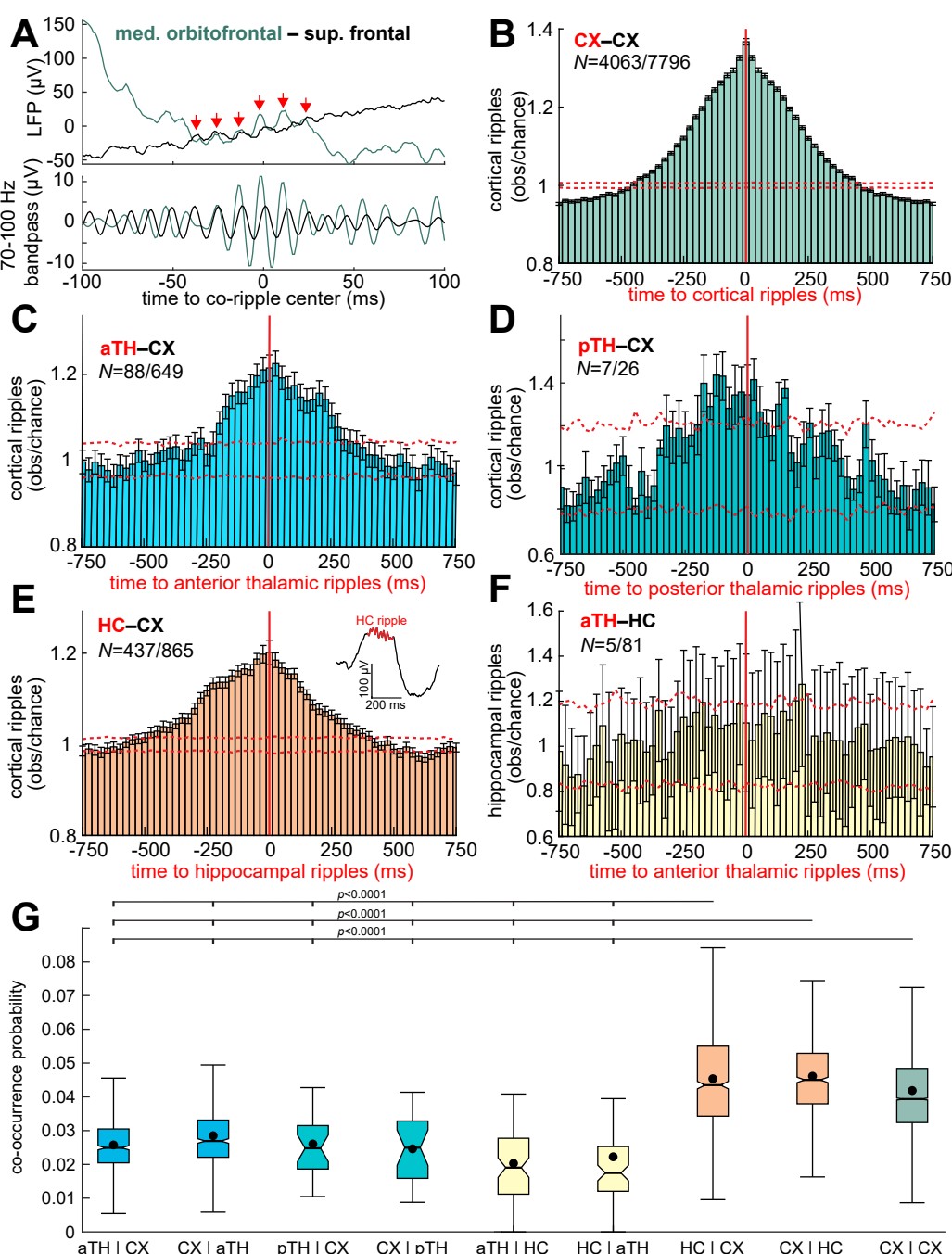

**Fig 3. Thalamic ripples infrequently and weakly co-occur with cortical and hippocampal ripples during NREM.** (**A**) Example co-occurring cortico-cortical ripples. (**B**) Average observed over chance cortical ripple centers in one channel relative to cortical ripple centers in another channel for significant channel pairs ($N$ = 4,063/7,796), sub-selected to demonstrate effect time courses. Note that no cortical bipolar channels included shared a common contact. Error bars show SEM and dashed error shows 98% confidence interval of the null distribution. See S11 Fig for results from all channel pairs. (**C**) Cortical ripples relative to anterior thalamic ripples ($N$ = 88/649 significant, post-FDR $p$ < 0.05, randomization test with shuffled controls; 25 ms Gaussian smoothed with $\sigma$ = 5 ms). (**D**) Same as C except for posterior thalamic ripples ($N$ = 7/26). (**E**) Same as B except hippocampal ripples relative to cortical ripples ($N$ = 437/865). Note the single sweep example of a hippocampal ripple on a sharpwave. (**F**) Same as B except hippocampal ripples relative to anterior thalamic ripples ($N$ = 5/81). (**G**) Conditional probabilities of ripple co-occurrences ($\geq$25 ms overlap) between channels (e.g., aTH | CX = probability of an anterior thalamic ripple co-occurring with a given cortical ripple). Post-FDR $p$-values, two-sided two-sample $t$ test. Source data are available in S3 Data.

Cortico-cortical and hippocampo-cortical sites also were more likely to co-ripple than thalamo-cortical and thalamo-hippocampal (Fig 3G). Among anterior thalamo-cortical channel pairs, 14% (88/649) were significantly coupled (post-FDR $p < 0.05$, randomization test) and 6% (40/649) had a significant number of co-occurrences above chance (post-FDR $p < 0.05$, randomization test). Among posterior thalamo-cortical channel pairs, 27% (7/26) were significantly coupled and 31% (8/26) significantly co-occurred. Among anterior thalamo-hippocampal pairs, even fewer, only 6% (5/81) were significantly coupled and 2% (2/81) had a significant number of co-occurrences. In contrast, among cortico-cortical pairs, 52% (4,063/7,796) were significantly coupled (including both permutations for each pair) and 47% (1,828/3,898) had a significant number of co-occurrences. Among hippocampo-cortical pairs, 51% (437/865) were significantly coupled and 15% (126/865) had significant co-occurrences. All of these coupling relationships except thalamo-hippocampal were significant across all channels from all patients (cortico-cortical: $p < 1 \times 10^{-25}$, $z = 257.1$, one-sided Wilcoxon signed-rank test; anterior thalamo-cortical: $p < 1 \times 10^{-25}$, $z = 40.1$; posterior thalamo-cortical: $p = 6 \times 10^{-23}$, $z = 9.8$; hippocampo-cortical: $p < 1 \times 10^{-25}$, $z = 92.9$; anterior thalamo-hippocampal $p = 0.58$, $z = -0.2$). Anterior thalamo-cortical co-ripple coupling was significantly less common than cortico-cortical ($p < 0.00001$, $\chi^2 = 383.1$, $df = 1$) or hippocampo-cortical ($p < 0.00001$, $\chi^2 = 88.1$, $df = 1$), which was also the case for co-occurrences (thalamo-cortical versus cortico-cortical: $p < 0.00001$, $\chi^2 = 381.4$, $df = 1$; thalamo-cortical versus hippocampo-cortical: $p < 0.00001$, $\chi^2 = 26.8$, $df = 1$). Across these 4 types of region pairs, the proportion of channel pairs with significant ripple coupling was significantly different ($p < 0.00001$, $\chi^2 = 438.73$, $df = 3$), as was the proportion of channel pairs with significant numbers of ripple co-occurrences above chance ($p < 0.00001$, $\chi^2 = 485.3$, $df = 3$).

There were also significantly higher average thalamo-cortical co-ripple conditional probabilities, i.e., P(CX | TH), when selecting for ripples with amplitudes in the top 20% versus bottom 20% within channels (anterior thalamo-cortical channel pairs: $0.0079 \pm 0.011$ (SD) versus $0.0054 \pm 0.0053$; $p = 1 \times 10^{-8}$, $t = 5.6$, $df = 648$, one-sided paired $t$ test; posterior thalamo-cortical: $0.0072 \pm 0.053$ versus $0.0045 \pm 0.0038$; $p = 0.022$, $t = 2.1$, $df = 25$). This shows that larger amplitude thalamic and cortical ripples preferentially co-occur; however, cortico-cortical ripple co-occurrence probabilities are 5–8x higher than anterior thalamo-cortical and 13–20x higher than posterior thalamo-cortical ripple co-occurrence probabilities. Therefore, cortico-cortical co-ripple coordination is much greater than thalamo-cortical, even when selecting for large amplitude ripples in the thalamus and cortex.

## Thalamic ripples do not phase-lock with cortical or hippocampal ripples

We previously showed that cortico-cortical but not hippocampo-cortical co-ripples often strongly phase-lock, suggesting a possible intracortical mechanism [6]. However, it is also possible that phase-locking between cortical locations is driven by a common input from thalamic ripples. To determine if channel pairs had phase-locked ripples, we computed phase-locking values (PLVs) [29], which measure phase consistency independent of amplitude, here using the 70 to 100 Hz phase across co-occurring ripples ($\geq 25$ ms overlap) at each time point for each channel pair ($\geq 40$ co-ripples required per pair). To compute the significance of PLV modulations, we compared the PLVs during co-ripples to those during randomly selected baseline periods preceding the co-ripples (see Methods). We found only 1 anterior thalamo-cortical site pair had significantly phase-locked co-ripples (S12A Fig; $N = 1/451$ significant channel pairs, post-FDR $p < 0.05$, randomization test), and no posterior thalamo-cortical site pairs had significantly phase-locked co-ripples (S12B Fig; $N = 0/20$). Consistent with our previous results, cortico-cortical pairs had significantly phase-locked co-ripples (S12C Fig; $N = 52/$

2,833 and S12D Fig; $N = 7/20$), specifically about 8× as many as thalamo-cortical pairs, and no hippocampo-cortical pairs had phase-locked co-ripples (S12E Fig; $N = 0/490$), nor did any thalamo-hippocampal pairs (S12F Fig; $N = 0/47$). Across these 4 types of region pairs, the proportion of phase-locked channel pairs was significantly different ($p = 0.001$, $\chi^2 = 16.2$, $df = 3$). Thus, these results support our hypothesis that networks of phase-locked cortico-cortical co-ripples are directly driven intracortically rather than from thalamic or hippocampal inputs.

## Thalamo-cortical and thalamo-hippocampal co-spindles phase-lock

Given that cortical ripples occur during cortical spindles and phase-lock between cortical sites and between the cortex and hippocampus, we hypothesized that the thalamus organizes these networks by projecting spindles and Up states to the cortex (Fig 4A). To test this hypothesis, we first computed 10 to 16 Hz PLVs between spindles in the thalamus and cortex. Since the

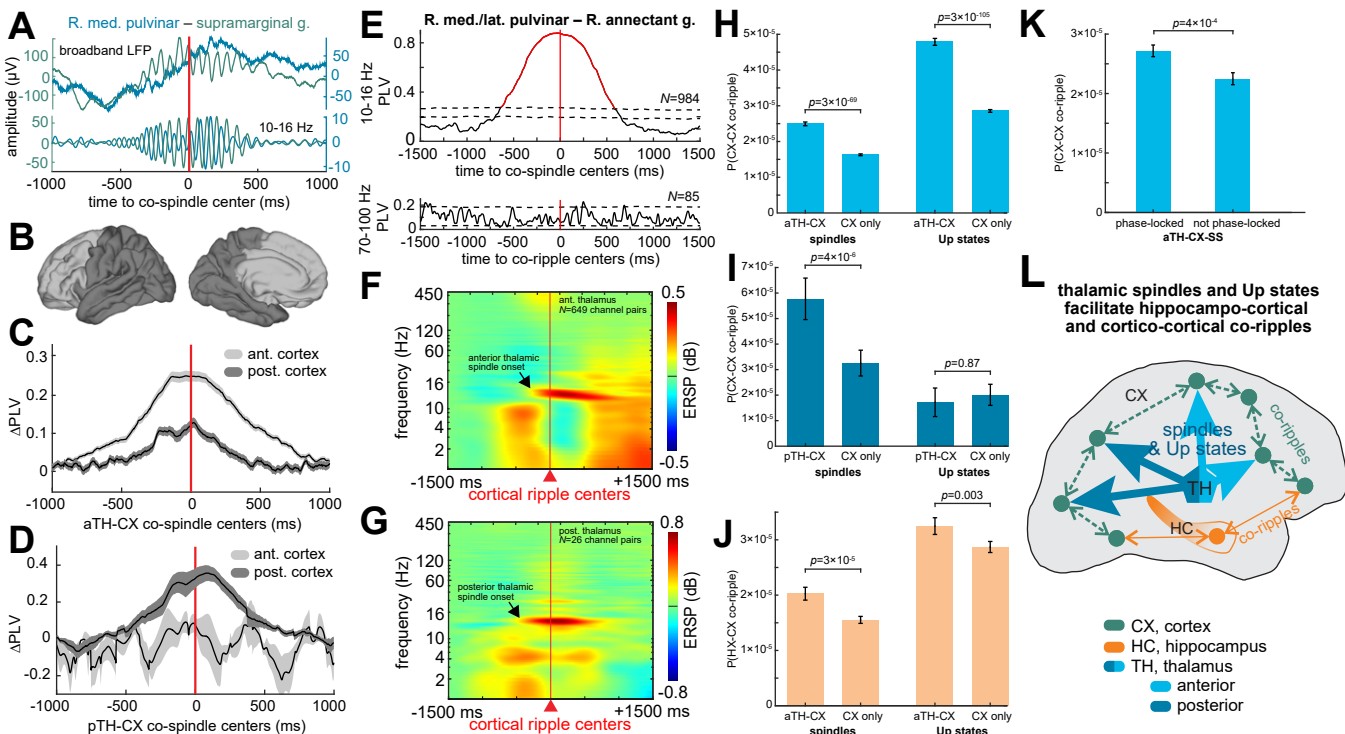

**Fig 4. Cortico-cortical and hippocampo-cortical co-rippling are enhanced during thalamo-cortical spindles and Up states.** (**A**) Example thalamo-cortical co-spindle in broadband and 10–16 Hz bandpass. (**B**) Division of the cortex into anterior and posterior regions based on connectivity with the anterior and posterior thalamus. (**C, D**) Mean and standard deviation 10–16 Hz PLVs of thalamo-cortical co-spindles, including anterior thalamus (**C**) and posterior thalamus (**D**), across channel pairs in anterior cortex ($N_{aTH} = 188/369$ and $N_{pTH} = 9/22$ significant, post-FDR $p < 0.05$, randomization test) compared to posterior cortex ($N_{aTH} = 71/280$ and $N_{pTH} = 0/4$). Note greater phase-locking between anterior thalamus with anterior cortex and between posterior thalamus with posterior cortex. Results by cortical parcel are shown in S13 Fig. (**E**) Example thalamo-cortical pair showing very high 10–16 Hz co-spindle phase-locking with no detectable 70–100 Hz co-ripple phase-locking. (**F, G**) Average time-frequency of anterior (**F**) and posterior (**G**) thalamus time-locked to cortical ripples ($N_{aTH} = 649$ and $N_{pTH} = 26$). (**H**) Cortico-cortical co-ripple probability across cortico-cortical channel pairs when the anterior thalamus is spindling with either cortical channel vs. when it is not (left bars; $p = 3 \times 10^{-69}$, $t(2915) = 18.0$; one-sided paired $t$ test), and similarly when the anterior thalamus and cortex share Up states vs. when they do not (right bars; $p = 3 \times 10^{-105}$, $t(2948) = 22.7$). (**I**) Same as H except posterior thalamus (spindles: $p = 4 \times 10^{-6}$, $t(17) = 6.2$; Up states: $p = 0.87$; $t(27) = 1.1$). (**J**) Same as H except hippocampo-cortical co-ripple probabilities (spindles: $p = 3 \times 10^{-5}$, $t(543) = 4.0$; Up states: $p = 0.003$, $t(544) = 2.7$). (**K**) Same as H except cortico-cortical co-ripple probability when one vs. neither cortical channel has 10–16 Hz phase-locked spindles with the anterior thalamus (in both cases TH and at least one of the CX channels are co-spindling; $p = 4 \times 10^{-4}$, $t(1937) = 3.3$). (**L**) Proposed synergistic mechanisms for achieving widely distributed cortical ripples that co-occur and phase-lock. Cortical depolarization via thalamo-cortical spindles and Up states (blue) trigger ripples which phase-lock through cortico-cortical coupling (green), and co-occur with hippocampal ripples via direct connections (orange). Our data do not support the alternative hypothesis wherein thalamic ripples directly drive cortical and/or hippocampal ripples. Source data are available in S4 Data.

anterior thalamus has more projections to the anterior than posterior cortex, we analyzed these regions separately (Fig 4B). In our analysis, the anterior cortex included orbitofrontal, prefrontal, and cingulate cortices, whereas the posterior cortex included all other cortical areas. We found significant thalamo-cortical spindle PLV modulations between anterior thalamus and anterior cortex (Fig 4C; $N = 188/369$ significant pairs, post-FDR $p < 0.05$, randomization test) as well as between anterior thalamus and posterior cortex ($N = 71/280$). There was a significantly greater proportion of significant thalamo-cortical channel pairs between anterior thalamus and anterior cortex compared to anterior thalamus and posterior cortex ($p < 0.00001$, $\chi^2 = 43.5$, $df = 1$). There was no significant thalamo-cortical spindle PLV between posterior thalamus and anterior cortex (Fig 4D; $N = 0/4$), but there was significant thalamo-cortical spindle PLV between posterior thalamus and posterior cortex ($N = 9/22$). Furthermore, there was a greater mean PLV modulation amplitude within ±250 ms around the thalamo-cortical temporal centers in the anterior compared to posterior cortex ($p = 5 \times 10^{-15}$, $t(613) = 7.9$, one-sided two-sample $t$ test). Results by individual cortical parcel are shown in S13 Fig. These results are consistent with preferential projections of the anterior thalamus to the anterior cortex and the posterior thalamus to the posterior cortex. Notably, our data show prominent phase-locking between thalamo-cortical spindles but not ripples (Figs 4C–4E, S12A, and S12B). We also evaluated 10 to 16 Hz phase-locking of spindles co-occurring between the anterior thalamus and hippocampus and found that 31/81 (38%) thalamo-hippocampal channel pairs had significantly phase-locked co-spindles.

## Cortico-cortical and hippocampo-cortical co-ripples couple to thalamo-cortical co-spindles and Up states

Because the data presented above do not support direct driving of cortical ripple co-occurrence by thalamic or hippocampal ripples, we hypothesized that they may be organized by interactions of slower waves, onto which ripples occur. This could be organized through the propagation of thalamic spindles to the cortex, which is supported by thalamic spindles preceding cortical spindles [15], as shown in S14A Fig. We found that cortical ripples tend to occur during thalamic spindles (Fig 4F and 4G; see S15 Fig for cortical activity relative to thalamic ripples), which was also the case for cortico-cortical co-ripples (S14B Fig). Furthermore, hippocampo-cortical co-ripples are also coordinated with thalamic spindle onsets (S14C Fig). The probability of a cortico-cortical co-ripple occurring during a cortical spindle (on either channel) was significantly increased by 53% when there was a co-occurring anterior thalamic spindle (on any channel) (Fig 4H; $p = 3 \times 10^{-69}$, $t(2915) = 18.0$, one-sided paired $t$ test). Likewise, the probability of a cortico-cortical co-ripple preceding a cortical Up state peak (within 500 ms) was significantly increased by 68% when there was a preceding thalamic Up state peak (within 500 ms from the cortical Up state peak) (Fig 4H; $p = 3 \times 10^{-105}$, $t(2948) = 22.7$). Thus, these data indicate that projecting thalamic spindles and Up states (but not ripples) help to coordinate cortico-cortical co-ripples (Fig 4L). Similarly, there was a 77% increase in cortico-cortical co-rippling during cortical spindles that co-occurred with a posterior thalamic spindle (Fig 4I; $p = 4 \times 10^{-6}$, $t(17) = 6.2$). However, there was no corresponding significant increase of cortical co-rippling associated with posterior thalamic Up states ($p = 0.87$, $t(27) = 1.1$).

In performing these same analyses except with respect to hippocampo-cortical co-ripples, the probability of a hippocampo-cortical co-ripple was significantly increased by 31% during anterior thalamo-cortical versus isolated cortical spindles (Fig 4J; $p = 3 \times 10^{-5}$, $t(543) = 4.0$), and significantly by 11% preceding a cortical Up state that was preceded by an anterior thalamic Up state (Fig 4J; $p = 0.003$, $t(544) = 2.7$).

### Cortico-cortical co-rippling is enhanced during phase-locked thalamo-cortical co-spindles

We showed that if a cortical site spindles with the thalamus during a ripple, it is more likely to co-ripple with another cortical site. We also found that thalamic spindles phase-lock with cortical spindles, consistent with a thalamo-cortical projection of spindles. We therefore hypothesized that cortico-cortical co-rippling is driven by phase-locked thalamo-cortical spindles. We tested if cortico-cortical co-ripple probability is greater between cortical sites that have 10 to 16 Hz spindles phase-locked with the thalamus versus those that do not. We found an increased cortico-cortical co-ripple probability when at least 1 cortical site had phase-locked spindles with the thalamus (Fig 4K; $p = 4 \times 10^{-4}$, $t(1936) = 3.3$). By contrast, there was no significant increase in hippocampo-cortical co-ripple probability among thalamo-hippocampal ($p = 0.86$, $t(772) = 1.1$), thalamo-cortical ($p = 0.83$, $t(662) = 0.97$), or both channel pairs ($p = 0.73$, $t(360) = 0.61$) that had phase-locked co-spindles versus those that did not. Thus, our data indicate that cortico-cortical co-rippling is further enhanced when thalamo-cortical spindles phase-lock.

## Discussion

Previous work has shown that human cortical ripples often co-occur and even phase-lock at long separations, posing the question of whether they are synchronized by a subcortical input [6]. The present study investigates if ripples also occur in the human thalamus, and if so whether they, or other thalamic waves, have a role in synchronizing cortical ripples. Indeed, we found that ripples occur in the human anterior and posterior thalamus during NREM, with similar oscillation frequency, occurrence density, and duration as neocortical and hippocampal ripples. However, thalamic ripples rarely co-occurred or phase-locked with cortical ripples, rendering them unlikely candidates for synchronizing ripples between widespread cortical locations. In contrast, thalamo-cortical spindles and Up states did appear to coordinate cortico-cortical and cortico-hippocampal ripple co-occurrence.

The term "ripple" was originally used to refer to a brief burst of oscillations at approximately 150 Hz near the peak of a much slower "sharpwave," recorded in the rodent hippocampus during NREM, and associated with memory replay [2]. Subsequently, similar oscillatory bursts were observed in different species, states, and structures, with different oscillation frequencies, and different behavioral and slower-wave contexts. Such oscillations are also often referred to generically as "ripples," with the recognition that their relationship to hippocampal sharpwave-ripples, and to each other, needs to be considered in each instance. Nonetheless, using a common terminology for such entities is a useful heuristic enabling observations in different species, states, structures and behaviors to be integrated.

The ripples we describe here are different from classic ripples in that they are ~90 Hz rather than ~150 Hz, associated with spindles and Up states rather than sharpwaves, and recorded in human thalamus rather than rodent hippocampus. However, experimental evidence supports their classification with classic ripples despite these differences. Putative ripples in humans were originally recorded in the hippocampus in several studies, as oscillations at ~90 Hz [30–32], as reviewed in Jiang and colleagues [33]. In hippocampi with no interictal spikes, these oscillations were recorded in NREM by contacts localized to stratum pyramidale of CA1, synchronized to waveforms resembling rodent sharpwaves recorded in adjacent contacts localized to stratum oriens [33]. Furthermore, these human hippocampal ripples were associated with cortical spindles and Up states [23,24]. These characteristics of putative human hippocampal ripples also characterize those in rodents [2], although some species differences may exist [34]. More recently, ~150 Hz cortical ripples were reported in rats, not associated with local

sharpwaves, but tending to co-occur with hippocampal sharpwave ripples, ripples in other cortical areas, and cortical spindles and Up states [22]. Like hippocampal sharpwave ripples, cortical unit spiking was phase-modulated by the local ripple, with putative pyramidal cell-firing slightly preceding putative interneurons. All of these same characteristics were reported for cortical 90 Hz oscillatory bursts in humans during NREM which were therefore also termed "ripples" [5,6,35]. We term the thalamic events described here "ripples" because they have similar frequency and duration as cortical and hippocampal ripples in humans, and tended to occur after the Down state, and those in the anterior thalamus tended to occur just before the subsequent Up state peak, at the onset of the local sleep spindle, like cortical ripples in humans.

The terms "high-gamma" and "high-frequency oscillation" are too generic to capture the stereotyped phenomenon that we report here since these terms lack specificity for frequency and duration, do not necessarily refer to discrete events, and may not require distinct oscillations or consistent focal frequencies. Within this broad and somewhat ill-defined range, there are multiple phenomena with different frequencies, generators, distributions and behavioral correlates (e.g., [36–38]. Consequently, we ensured that our detected events were not only in a specific frequency range, and larger than surrounding activity in that range for a minimum duration, but that each consisted of multiple waves with similar amplitude and duration, i.e., that they are true oscillations at approximately 90 Hz (see Methods). These characteristics may be crucial for the generation and function of co-ripples. Specifically relevant to the current results, co-occurrence of non-oscillatory cortical gamma above 150 Hz fails to exhibit the strong task-modulation shown by cortical ripple co-occurrence [10]. Thus, the term "ripple" aptly applies to the thalamic events here, provided that one broadens the term "ripple" to include events in other sites and states, and presumably with a broader function. In the current study, as in most other studies of human hippocampal ripples, we did not restrict hippocampal ripples to those associated with sharpwaves, in order to maximize the detection of co-ripples with the thalamus or cortex.

Gamma generation in the thalamus and transmission to the cortex have previously been studied. Thalamic gamma oscillations, usually at 40 Hz but extending to 80 Hz, have been recorded in cats and rodents, where they are generated by voltage-gated dendritic $Ca^{2+}$ currents [39]. This rhythmic firing led to the proposal that the thalamus drives synchronous gamma oscillations across widespread neocortical areas via the widely distributed thalamo-cortical projections of the matrix system [39–41], but direct evidence is absent. Gamma oscillations have also been reported in mouse and primate visual cortex [42]. However, these are also lower frequency events (approximately 50 Hz), which appear to have a more restricted anatomical distribution and only synchronize over short distances [43], and may therefore represent a distinct phenomenon.

While it is impossible to definitively exclude that our results are not influenced by epilepsy, we carefully sub-selected patients, channels, epochs, and events using iteratively developed manual and automatic methods, in order to exclude epileptic activity. Furthermore, the ripples we detected have the same fundamental characteristics of those detected in microelectrode recordings from the cortex of patients with tetraplegia but not epilepsy [9,44].

A direct wave-by-wave driving of cortical ripples by thalamic ripples would imply strong synchrony between them. However, thalamic ripples only weakly co-occurred and rarely phase-locked with cortical ripples. These findings, in stark contrast to the robust ripple co-occurrence between cortical sites in this and our previous study [6], suggest that thalamic ripples do not directly drive widespread forebrain ripple synchrony. Although cortical ripples did not synchronize with thalamic ripples, cortico-cortical co-rippling was strongly promoted by the co-occurrence of spindles or Up states in the thalamus and cortex. Critically, when spindles

or Up states coincided between the thalamus and a cortical site, the cortical site was approximately 40% to 50% more likely to co-ripple with another cortical site. Furthermore, spindles preferentially phase-locked between anterior thalamus and anterior cortex, and between posterior thalamus and posterior cortex, as also found in Mak-McCully and colleagues [15], which is consistent with anatomical projections from the thalamus. We found that thalamo-cortical spindle phase-locking further enhanced cortico-cortical co-rippling, but neither thalamo-hippocampal nor thalamo-cortical spindle phase-locking enhanced hippocampo-cortical co-rippling. Thus, the thalamus appears to coordinate cortical co-rippling, not by projecting ripples, but by projecting synchronous spindles and Up states. This is consistent with the functional coupling between thalamic and cortical spindles and Down-to-Up states in humans [15,27] and animals [20,21].

Thalamic spindles and Up states may trigger cortical ripples by depolarizing cortical neurons for approximately 50 to 100 ms. Evidence for such depolarizations during NREM has been found in the prominent current sinks and large increases in pyramidal and interneuron firing in microelectrode array recordings from human cortex during Up states and spindles [5,18,19,45]. Rodent hippocampal ripples can be induced by such broad approximately 60 ms duration depolarizing pulses, via a mechanism involving pyramidal-basket cell feedback loops ("PING") synchronized by basket–basket cell co-firing ("ING") [46]. Further evidence that this mechanism underlies human cortical ripples during NREM is that they are accompanied by strongly phase-locked unit firing, with pyramidal cells leading interneurons [5].

Previous studies in humans suggest that hippocampal ripples facilitate information transfer to the cortex during sleep [23–25,47]. Both anterior and posterior thalamic spindles co-occur with cortical spindles [15,48]. We examined the three-way relationships among thalamic, hippocampal, and cortical ripples, and found that thalamic ripples only rarely and weakly co-occur and do not phase-lock with hippocampal ripples. Using new data, we also confirmed our previous findings that cortical ripples have limited co-occurrences and do not phase-lock with hippocampal ripples [6]. Thus, hippocampal ripples do not appear to be driven by thalamic ripples. However, since hippocampal ripples coordinate with cortical spindles, Down states, and Up states [23–25], the thalamus could coordinate ripples in the hippocampus and cortex, via a depolarizing modulation as outlined above, rather than via high-frequency oscillatory driving. Indeed, thalamic Up states that project to the cortex or hippocampus were associated with an increased probability of hippocampo-cortical co-ripples.

The thalamus is comprised of many nuclei subserving different physiological functions [27,48]. The present study distinguishes anterior and posterior thalamus, but further granularity was not possible in these rare recordings with limited sampling and relatively wide spacing between relatively large recording contacts. Therefore, our results only portray general relationships between the anterior and posterior thalamus with the cortex.

Overall, our findings are consistent with cortico-cortical ripple phase-locking arising from direct interactions between cortical neurons. Cortico-cortical ripple synchrony has been proposed to support the binding of specific information encoded in the interacting co-rippling locations [6,49]. In such models, not only are oscillations modulated by the interaction of cortical areas through phase-selection [50] and coincidence detection [51], but in addition these interactions themselves reinforce the oscillations through re-entrance [52–54]. A mutually reinforcing mechanism is consistent with our previous finding of a strong dependence of phase-locking on the co-rippling of several cortical locations simultaneously, beyond the 2 whose ripple phase-locking is being measured [6]. The synchronization of co-ripples appears to entrain reciprocal co-firing of neurons between areas, further enhanced by local Up states, supporting re-entrance [44]. Because the integration of specific information during binding requires high bandwidth, it may be most effectively transmitted by direct cortico-cortical

connections in humans, given that neuroanatomical calculations imply that >95% of synapses on cortical cells are from the cortex (reviewed in Rosen and Halgren [55]). In particular, the ratio of cortical neurons to thalamo-cortical projection cells is approximately 1,400:1, based on human thalamic cell counts in Xuereb and colleagues [56], cortical cell counts in Azevedo and colleagues [57], and the proportion of thalamo-cortical cells in Arcelli and colleagues [58]. Thus, phase-locking arising from cortico-cortical interactions is consistent with the proposed role of co-ripples in supporting cross-cortical information binding. Our finding that thalamo-cortical spindles and Up states, but not thalamic ripples, enhance cortico-cortical and hippo-campo-cortical co-rippling is consistent with the thalamus having a slower, modulatory rather than a high-frequency, direct driving role in promoting co-rippling.

In sum, we provide the first evidence of ripples in the human anterior and posterior thalamus with similar characteristics as hippocampal and cortical ripples. Due to infrequent and weak co-occurrence and phase-locking between ripples in the thalamus and those in the cortex or hippocampus, it is unlikely that thalamic ripples directly drive cortico-cortical or hippo-campo-cortical ripple co-occurrence or phase-locking. Rather, we found evidence that thalamic Up states, spindles, and especially phase-locked spindles coordinate the co-occurrence of ripples in the cortex. This may help to organize replay that underlies the consolidation of memories across widespread areas in the cortex.

## Methods

### Ethics statement

All patients in the study gave fully informed written consent for their data to be used for research. This study was approved by the local Institutional Review Boards at the University of Alabama, Birmingham (patients 1 to 10 with anterior thalamic electrodes) and the French Institute of Health (patients 11 to 13 with posterior thalamic electrodes) (S1 Table), in accordance with the principles of the Declaration of Helsinki.

### Patient selection, intracranial recordings, and data preprocessing

Data from 13 patients (8 female, 39.5 ± 11.5 years old) with pharmaco-resistant focal epilepsy undergoing intracranial recording for approximately 1 to 2 weeks to localize seizure onset zones prior to surgical resection were included in this study (S1 Table). None of these data were included in our previous studies on cortical ripples [5,6,10,44]. The patients included in this study underwent extensive clinical noninvasive evaluation that indicated that they had medication-resistant focal epilepsy that could be treated surgically. However, it is difficult to determine the location of the epileptic foci noninvasively, and these foci vary across patients. Therefore, electrode targeting was performed in a patient-specific manner in order to target each patient's suspected epileptic foci.

As reviewed by Gadot and colleagues [59], implantation of the thalamus together with cortical sites is a clinically accepted procedure in the presurgical evaluation of partial epilepsy. Partial epilepsy is typically a network phenomenon, and the thalamus is the most common extra-temporal site that is recruited by temporal lobe seizures [60–62]. Thus, thalamic recordings can provide important information regarding the sequence of engagement of different cortical areas during the evolution of the seizure discharge. Although the thalamus is not a target for resection, it can be a target for FDA-approved deep brain stimulation therapy. Through an IRB-approved process, we have prospectively recruited adults (>18 years old) undergoing SEEG investigation for the localization of seizure foci to participate in the thalamic electrophysiological recordings. Fully written informed consent was obtained from all patients before the thalamic SEEG implantation. Implantation and consent procedures for the patients

with anterior thalamic electrodes have been described in detail in our prior study [63]. Those procedures for the posterior thalamic electrodes were highly similar. In summary, one of the clinically indicated depth electrodes sampling the insula operculum was advanced deeper to target thalamic nuclei. Thus, we avoided using additional depth electrodes for exclusive research implantation and mitigated the risk of bleeding, which is known to be higher with an increased number of depth electrodes implanted. None of the patients had a thalamic bleed.

Patients were only included in this study if they had no prior brain surgery and normal background LFPs except for infrequent epileptiform activity. A total of 15 patients with thalamic electrodes were rejected due to abnormal LFPs if they met any of the following criteria: (1) more than 1 automatically detected (see below) putative interictal spike per minute on average; (2) more than one >1 mV artifact per minute on average as determined visually across ≥60 min of data. Patients 1 to 10 recordings were collected from PMT electrodes (PMT Corporation, Chanhassen, Minnesota, USA) with a Natus Quantum amplifier (Natus Medical Incorporated, Pleasanton, California, USA) at 2,048 Hz sampling with a 0.016 to 683 Hz bandpass. Patient 11 recordings were collected at 512 Hz with a 160 Hz bandpass. Patients 12 and 13 recordings from DIXI electrodes (DIXI Medical, Marchaux-Chaudefontaine, France) were collected at 1,024 Hz with a 0.16 to 340 Hz bandpass. PMT electrodes were 0.8 mm diameter with 10 to 16 2 mm long contacts separated by 1.5 to 3 mm (approximately 150 contacts/patient). Data preprocessing was performed in Matlab 2019b (MathWorks, Natick, Massachusetts, USA) and LFPs were inspected visually using the Fieldtrip toolbox [64]. Data sampled above 1,000 Hz were downsampled to 1,000 Hz with anti-aliasing at 500 Hz. All data were notch filtered at 60 Hz with 60 Hz harmonics (patients 1 to 10) or 50 Hz with 50 Hz harmonics (patients 11 to 13) up to the Nyquist frequency.

## Channel selection

Channels were bipolar derivations of adjacent contacts to ensure measurement of LFPs. Channels were only included if their contacts were in non-epileptogenic, non-lesioned thalamus as well as cortex, and in 9 of the 13 patients also hippocampus. Furthermore, cortical and hippocampal channels were excluded if they shared a common contact with another bipolar channel that was included in the analysis, whereas all thalamic bipolars that met criteria were included in order to maximize sampling from the many nuclei in the thalamus. In order to ensure that recordings reflect local activity rather than volume conducted from elsewhere, we restricted recordings to the gray matter. Initial selection of bipolar contacts that may be in the gray matter was made by using the co-registered preoperative MRI and postoperative CT to localize contacts. However, the cortical ribbon is thin (approximately 3 mm in humans), so in addition to using the co-registered preoperative MRI and postoperative CT to localize contacts, we used physiological measurements to select among possible disjoint bipolar pairs those with the largest spontaneous LFP amplitude. Specifically, channels were manually selected based on NREM averages of narrowband delta (0.5 to 2 Hz) and high-gamma (70 to 190 Hz) analytic amplitudes. Cortical channels selected had an average delta peak amplitude greater than 40 μV, hippocampal channels had a delta amplitude greater than 15 μV, and thalamic channels had a delta amplitude greater than 5 μV. In addition, they had significant correlations ($p < 0.05$) between the low-pass delta waveform and the high-gamma (70 to 190 Hz) analytic amplitude. Delta was used because its power is increased during NREM that is characterized by slower frequency oscillations including the slow oscillation and K-complexes. High gamma was used because its power reflects local neuron spiking generated by cells in the gray matter. Both are present in virtually all cortical and hippocampal bipolar channels during NREM stage N3. Channels were then visually inspected during NREM periods to ensure they did not have

frequent interictal spikes or artifacts and had a normal appearing broadband signal. Among 2,145 bipolar channels (1,157 left sided) across the patients in the study, a total of 336 channels (110 left sided) were included in the study. Among the 39 thalamic bipolar channels across the patients in the study, 30 channels were included in the study because they had normal appearing LFPs without evidence of frequent interictal spiking. Note that the majority of channels were excluded because they were not in thalamic, cortical, or hippocampal gray matter, and thus would not record meaningful local neuronal activity, or in order to keep bipolar derivations disjoint.

## Channel localization

Cortical surfaces were reconstructed using the preoperative T1-weighted structural MR volume with the FreeSurfer recon-all pipeline [65]. Atlas-based automated parcellation [66] was utilized to assign anatomical labels to cortical surface regions according to the Destrieux and colleagues [67] atlas and subsequently amalgamated into cortical parcels per Desikan and colleagues [68]. For analyses involving anterior versus posterior cortex, anterior cortex included Desikan parcels that comprise the orbitofrontal cortex (frontal pole, lateral orbitofrontal, medial orbitofrontal, pars orbitalis), prefrontal cortex (caudal middle frontal, pars opercularis, pars triangularis, rostral middle frontal, superior frontal), and cingulate cortex (caudal anterior cingulate, rostral anterior cingulate, posterior cingulate). Posterior cortex was comprised of the remaining cortical parcels.

To localize the SEEG contacts, the post-implant CT volume was co-registered to the pre-implant MR volume in standardized 1 mm isotropic FreeSurfer space with the general registration module [69] in 3D Slicer [70]. Next, each contact's position in FreeSurfer coordinates was determined by manually marking the contact centroid as visualized in the co-registered CT volume. Each transcortical bipolar channel was assigned an anatomical parcel from the above atlas through determining the parcel identities of the white-gray surface vertex nearest to the midpoint of the adjacent contacts. Automated segmentation assigned a nucleus to each voxel of the MR volume in the thalamus [71], which was confirmed by visual comparison with the Allen Brain Atlas (https://atlas.brain-map.org/). The 24 thalamic bipolar channels included the following nuclei (in descending frequency): AnteroVentral, Ventral Anterior, Reticular, Ventral Lateral, Fasciculus, MedioDorsal, and CentroMedian (S1 Table). Due to possible inaccuracies in registration and limitations of visible nuclear boundaries, assignment of contacts to particular nuclei should be considered approximate. Transcortical bipolar locations were registered to the fsaverage template brain for visualization through spherical morphing [72]. All contacts marked as thalamic or hippocampal were confirmed to localize to these respective structures through visual inspection of the co-registered preoperative MRI and postoperative CT. The posterior limit of the uncal head was used as a boundary to define anterior versus posterior hippocampus [73,74].

## Sleep staging and epoch selection

Sleep staging was performed to select NREM stages N2 and N3 consistent with Silber and colleagues [75]. Data were bandpassed at 0.5 to 2 Hz and then Hilbert transformed to obtain the narrowband delta analytic amplitudes. NREM epochs were selected for analysis when the cortical channels had increased delta analytic amplitude during overnight hours (8:00 PM to 8:00 AM). These epochs subsequently underwent visual confirmation that they contained frequent and prominent slow wave oscillations and spindles characteristic of NREM stages N2 and N3. Epochs were only retained when they were absent of frequent interictal spikes or artifacts. Notably, disruptions to sleep (e.g., for vitals assessment) may occur while patients are on the

epilepsy monitoring unit despite our methods being designed to select for the most normal overnight NREM in these patients.

## Time-frequency analyses

Time-frequency plots of ripple-triggered spectral power were generated from the broadband LFP using EEGLAB [76]. Power was calculated from 1 to 500 Hz by averaging the fast Fourier transforms with Hanning window tapering locked to the centers of 5,000 randomly sub-selected ripples per channel. Each 1 Hz frequency bin was normalized to the mean power at −2 to −1.5 s and then masked with two-tailed bootstrapped significance ($N$ = 200 shuffles) with $p$-values FDR-corrected with α = 0.05 using a −2 to −1.5 s baseline. Grand average time-frequency plots were produced by averaging across the channel average time-frequency plots.

## Ripple detection

Ripples were detected in the thalamus, cortex, and hippocampus based on previously published methods [5,6]. For each channel, data were bandpassed at 70 to 100 Hz with a sixth order forward-reverse Butterworth filter (zero-phase shift) and peaks in the analytic amplitude of the Hilbert transform were selected if they were at least 3 standard deviations above the channel mean. Event onsets and offsets were found on both sides of the peak when the 70 to 100 Hz analytic amplitude decreased below a z-score of 0.75. Events were retained if they had at least 3 oscillation peaks in the 120 Hz low pass, determined by moving a 40 ms window in 5 ms increments ±50 ms relative to the midpoint of the ripple, requiring at least 1 window to have at least 3 peaks. Adjacent events within 25 ms were merged and ripple centers were determined as the time of the largest positive peak in the 70 to 100 Hz bandpass.

To exclude epileptiform activity and artifacts, events were rejected when the absolute value of the z-score of the 100 Hz highpass was greater than 7. Events were also rejected if they were within ±500 ms from putative interictal spikes (detection criteria described below). Furthermore, events were rejected if they coincided with a putative interictal spike detected on any channel to exclude events that could be coupled across channels due to epileptiform activity. Lastly, events were rejected if the largest peak-to-valley or valley-to-peak absolute amplitude in the broadband LFP was 2.5 times greater than the third largest in order to exclude events that had only 1 prominent cycle or deflection. The average broadband LFP and average time-frequency plots locked to ripple centers as well as multiple individual broadband LFP ripples were visually examined in each channel from each patient to confirm there were multiple distinct cycles in the 70 to 100 Hz band, without contamination by possible artifacts or epileptiform transients. The oscillation frequency of each ripple was computed as follows:

$$f = \frac{N}{2 \times d} \tag{1}$$

Where $N$ is the number of 70 to 100 Hz zero crossings (i.e., half cycles, with fractional cycles determined based on the remaining phase angle over π), and $d$ is the duration of the ripple.

## Interictal spike rejection

Ripples were excluded from analysis if they were within 500 ms from putative interictal spikes. Putative interictal spikes were detected by finding a high frequency score, which was the 20 ms boxcar smoothed 70 to 190 Hz analytic amplitude, and a spike template score, which was computed as the cross-covariance with a template interictal spike. A putative interictal spike was marked when the sum of the high frequency score, weighted by 13, and the spike template score, weighted by 25, exceeded 130. These thresholds were determined through extensive

visual examination of many hundreds of events across multiple channels from multiple patients.

## Down state, Up state, and spindle detection

Down states and Up states were detected according to previously published methods [5,23,24,77]. Data were bandpassed from 0.1 to 4 Hz (delta) and the top 10% of amplitude peaks between consecutive zero crossings within 0.25 to 3 s were identified. The polarity of each bipolar signal was inverted if needed so that Down states were negative and Up states were positive. This was done by confirming that the average 70 to 190 Hz analytic amplitude ±100 ms (cortex) or ±250 ms (thalamus) around peaks was greater for Up states than Down states for each channel. Broadband high gamma (70 to 190 Hz) was used as it is a proxy for multi-unit activity that is enhanced during the Up state and silenced during the Down state. Since the high gamma bandpass of 70 to 190 Hz overlaps with the ripple detection band of 70 to 100 Hz, we also evaluated channel polarities using high gamma within 110 to 160 Hz, to avoid the ripple band and ripple oscillation frequency harmonics. We found that all thalamic channels and 98% of cortical channels had the same assigned polarities when using 110 to 160 Hz versus 70 to 190 Hz to assign polarities. Further determination of thalamic polarities, which may be difficult given the small magnitude of high gamma [15], was performed by visually inspecting the spindle-locked delta signal with respect to the high-gamma analytic amplitude. Polarity was determined such that the high-gamma envelope increased during the rising phase of the delta signal (i.e., the Down-to-Up state transition).

Spindles were also detected according to a previously published method [5,45,77]. Data were bandpassed at 10 to 16 Hz, and the absolute values were smoothed with convolution using a tapered 300 ms Tukey window, then median values were subtracted from each channel. Data were normalized by the median absolute deviation, and spindles were identified when peaks exceeded 1 for a minimum of 400 ms. Onsets and offsets were marked when these amplitudes fell below 1. Putative spindles coinciding with large increases in lower (4 to 8 Hz) or higher (18 to 25 Hz) power were rejected to exclude broadband events and theta bursts, which may overlap with slower spindle frequencies [16].

## Ripple timing and co-occurrence analyses

Timing relationships between ripples and other sleep waves were computed as previously described [5]. Times of ripple centers relative to spindle onsets, Down state peaks, and Up state peaks on the same channel were determined for each channel. Event counts were calculated in 50 ms bins ±1,500 ms around cortical ripple centers. Time histogram plots were smoothed with a 50 ms Gaussian window with σ = 10 ms.

Cortical ripples were considered to co-occur ("co-ripple") if they overlapped for at least 25 ms in 2 different cortical bipolar channels (no cortical bipolar channels shared a common contact). A cortical co-ripple was determined to co-occur with an isolated cortical spindle if its time center occurred during a spindle on either cortical channel without a thalamic spindle onset preceding the cortical spindle onset within 500 ms on any thalamic channel. A cortical co-ripple was determined to co-occur with a thalamo-cortical spindle if there was a thalamic spindle on any thalamic channel with an onset within 500 ms preceding the cortical spindle onset. Probabilities were computed as the number of co-ripples divided by the total time of cortical spindling, including only the cortical spindles selected for each condition as described above. This same method was used to compute the probability that a cortical co-ripple occurred within 500 ms preceding isolated cortical Up states compared to thalamo-cortical Up states, where the thalamic Up state peak preceded the cortical Up state peak within 500 ms.

To compute the probability of cortico-cortical (CX1-CX2) co-rippling given cortical (CX3) rippling, we divided the instantaneous probability of CX1-CX2-CX3 co-rippling by the instantaneous probability of CX3 rippling. Likewise, to compute the probability of CX1-CX2 co-rippling given thalamic (TH) rippling, we divided the instantaneous probability of TH-CX1-CX2 co-rippling by the instantaneous probability of TH rippling. For each CX1-CX2 pair, CX3 and TH was a different, randomly selected channel. Co-rippling required ≥25 ms overlap.

## Phase-locking analyses

Phase-locking of co-occurring ripples/spindles at different sites was evaluated using the PLV, which is an instantaneous measure of phase consistency that is independent of signal amplitudes [29]. PLV time courses were determined using the analytic angles of the 70 to 100 Hz (for ripples) or 10 to 16 Hz (for spindles) bandpass (sixth order zero-phase shift forward-reverse filtered) of each channel pair when there were at least 40 co-ripples that overlapped for at least 25 ms or 40 co-spindles that overlapped for at least 200 ms. PLVs were computed in 1 ms timesteps relative to the temporal centers of the co-ripples/spindles. A null distribution was generated by randomly selecting 500 times within −10 to −2 s preceding each co-ripple/spindle center. Pre-FDR $p$-values were computed by comparing the observed versus null distributions in 5 ms bins within ±50 ms around the co-ripple centers or within 50 ms bins within ±250 ms around the co-spindle centers. $P$-values were then FDR-corrected across bins and channel pairs. A channel pair was determined to have significant ripple phase-locking if at least 2 consecutive bins had post-FDR $p < 0.05$. PLV plots were Gaussian smoothed with a window of 10 ms and σ = 2 ms.

## White matter controls

Bipolar channels, recorded from the electrodes implanted into the thalamus, which localized to the white matter lateral to the thalamic bipolar channels included in the study, were selected for control analyses. One white matter bipolar channel that met these criteria was identified in patients 2, 3, 4, 7, and 10. To test for effects of volume conduction, the mean white matter channel LFP relative to thalamic ripple centers was computed. To test for ripples in the white matter channels, ripple detection was performed with the same criteria as described above using the thalamic channel thresholds.

## Experimental design and statistical analyses

Statistical analyses were performed with $\alpha$ = 0.05 and all $p$-values involving multiple comparisons were FDR-corrected [78] across all channels from all patients unless specified otherwise. Random shuffling statistics were performed with $N$ = 200 iterations per channel or channel pair. Paired and two-sample $t$ tests were used to compare distribution means. To determine if a peri-event time histogram was significant for a given channel or channel pair, a null distribution was found by shuffling the events relative to the ripples within ±1.5 s 200 times per channel pair or channel. $P$-values were computed by a comparison of the observed and null distributions for each bin over ±25 ms for ripples or ±50 ms (250 ms Gaussian smoothed with σ = 50 ms) for other sleep waves. All channel pairs from all patients were included regardless of the numbers of events (co-occurring or independently) for these pair-wise statistical analyses. These $p$-values were then FDR-corrected across all bins across all channels/channel pairs. A channel/channel pair had a significant modulation if 3 or more consecutive bins had post-FDR $p < 0.05$. A one-sided Wilcoxon signed-rank test was used to test for significance of modulation across all channels or channel pairs using the observed versus mean null distribution bin values within −500 to 0 ms and 0 to 500 ms or −500 to 500 ms, respectively. Whether a

given channel/channel pair had a leading/lagging relationship between events was determined using a two-sided binomial test to compare the counts in the 250 ms preceding versus following $t = 0$ with an expected value of 0.5. Whether a channel pair had significantly greater co-rippling compared to chance was determined by randomly shuffling ($N = 200$ iterations) the inter-ripple ripple intervals in moving non-overlapping 5 min windows on each channel and comparing the observed versus null number of co-occurrences. These $p$-values were FDR-corrected across all channel pairs. Conditional probabilities of cortico-cortical relationships were computed for both P(A|B) and P(B|A). Comparisons of proportions (e.g., proportions of significant channel pairs for various conditions) were performed with a $\chi^2$ test of proportions. For a given pair-wise analysis, e.g., cortico-cortical ripple co-occurrences, the strength of the relationship was considered to be weak if increased by <25% from chance, or was otherwise considered to be strong. Likewise, the relationship was considered to be infrequent if <25% of channels were significant, or was otherwise considered to be frequent. For analyses of spindle- and Up state-associated changes in co-rippling, channel pairs were excluded from statistical analysis if there were no co-ripples in either condition being compared or if there was less than 15 s of total cortical spindling time.

Ripple characteristics were plotted using Violinplot-Matlab (https://github.com/bastibe/Violinplot-Matlab). To test for differences of ripple characteristics (i.e., density, amplitude, frequency, duration) between different regions, we used the following linear-mixed effects model formula to control for patient as a random effect, with subsequent FDR-correction for multiple comparisons (across all channels from all patients):

$$characteristic \sim region + (1|patient). \tag{2}$$

## Supporting information

**S1 Fig. Example thalamic contact localizations.** Preoperative MR axial section in grayscale with co-registered post-operative CT showing left sided thalamic contacts in orange from a representative patient. Overlaid outlines show thalamic nuclei as estimated by automated segmentation of the T1-weighted MR volume [71]. CT, computed tomography; MR, magnetic resonance. See S1 Table for thalamic channel localizations of each patient.
(TIF)

**S2 Fig. Cortical channel maps.** Markers show centers of all bipolar channels ($N = 275$) from all patients ($N = 13$). Further anatomical information, including thalamic and hippocampal localizations, are provided in S1 Table.
(TIF)

**S3 Fig. Detection of thalamic ripples is not due to volume conduction.** Average and SEM thalamic LFP (left column) and white matter LFP (right column) time-locked to thalamic ripples. All recordings are bipolar referenced in order to ensure focal measurement of LFPs. Note the prominent ripple oscillation that localizes to thalamic gray matter but is not present in the adjacent white matter. LFP, local field potential; SEM, standard error of the mean.
(TIF)

**S4 Fig. Ripples are almost never detected in white matter adjacent to the thalamus.** Density (frequency of occurrence) of events detected in bipolar channels that localize to the white matter adjacent to the thalamus. Recordings were obtained from the same probes with medial contacts implanted into the thalamus. The ripple densities in the white matter were 4% of those in the cortex as reported in Fig 1E ($p = 5 \times 10^{-10}$, $t(27) = 9.4$; linear mixed-effects with patient as

random effect), indicating that the ripples included in this study localize to gray matter and are not due to volume conduction, noise, or artifact. Horizontal lines, means; circles, medians; vertical lines, interquartile ranges. Source data are available in S5 Data.
(TIF)

**S5 Fig. Characteristics of cortical ripples from patients with posterior thalamic recordings.** (**A–D**) Cortical ripple densities (**A**), peak 70–100 Hz analytic amplitudes (**B**), oscillation frequencies (**C**), and durations (**D**) during NREM across all channels ($N = 14$ from patients 11–13). Horizontal lines, means; circles, medians; vertical lines, interquartile ranges. CX, cortex; NREM, non-rapid eye movement sleep. Source data are available in S6 Data.
(TIF)

**S6 Fig. Thalamic, cortical, and hippocampal ripple characteristics across individual ripples. (A–D)** Anterior thalamic (**A**), posterior thalamic (**B**), cortical (**C**), and hippocampal (**D**) peak 70–100 Hz analytic amplitudes, oscillation frequencies, and durations across all ripples ($N_{aTH} = 88{,}320$, $N_{pTH} = 8{,}996$, $N_{CX} = 857{,}420$, $N_{HC} = 135{,}646$) during NREM. Values above each plot are mean and standard deviation. Horizontal lines, means; circles, medians; vertical lines, interquartile ranges. aTH, anterior thalamus; CX, cortex; HC, hippocampus; NREM, non-rapid eye movement sleep; pTH, posterior thalamus. Source data are available in S7 Data.
(TIF)

**S7 Fig. Thalamic ripple characteristics by patient. (A–D)** Thalamic ripple densities (**A**), peak 70–100 Hz analytic amplitudes (**B**), oscillation frequencies (**C**), and durations (**D**) across ripples from each patient. Patients 1–10 are anterior thalamus and 11–13 are posterior thalamus. Boxes show interquartile ranges, horizontal lines indicate medians, and whiskers represent 1.5 × interquartile range. Source data are available in S8 Data.
(TIF)

**S8 Fig. Thalamic ripples occur on the local Down-to-Up state transition during spindles.** (**A**) Average and SEM times of anterior thalamic Down state peaks relative to anterior thalamic ripples. (**B**) Same as A except posterior thalamus. (**C, D**) Same as A and B except spindle onsets. Shaded boxes denote average spindle interval. (**E, F**) Same as A and B except Up state peaks. Data are from all channels from all patients. Channels exclusively with significant modulations are depicted in Fig 2. Dashed errors show 98% confidence intervals of the null distribution. *P*-values were computed using a Wilcoxon ranked-sum test to compare the modulation amplitude within −500 to 0 ms and 0 to 500 ms across bins and channels for observed values vs. null mean values. ns = nonsignificant, *$p < 0.05$, **$p < 0.01$, ****$p < 0.0001$. Source data are available in S9 Data.
(TIF)

**S9 Fig. Thalamic ripples may occur during cortical spindles on the Down-to-Up state transition.** (**A**) Average and SEM times of cortical Down state peaks relative to anterior thalamic ripples (anterior thalamus: $N_{aTH} = 36/649$ channel pairs significant, post-FDR $p < 0.05$, randomization test with shuffled controls). (**B**) Same as A except posterior thalamus ($N_{pTH} = 3/26$). (**C, D**) Same as A and B except cortical spindle onsets ($N_{aTH} = 70/649$, $N_{pTH}$ 7/26). (**E, F**) Same as A and B except cortical Up state peaks ($N_{aTH} = 31/649$; $N_{pTH} = 5/26$). Data are from all channels from all patients. Dashed errors show 98% confidence intervals of the null distribution. aTH = anterior thalamus; FDR, false discovery rate; pTH = posterior thalamus. Source data are available in S10 Data.
(TIF)

**S10 Fig. Thalamic ripple autocorrelation.** (**A, B**) Average and SEM within-channel autocorrelation of anterior (**A**; $N$ = 24 channels) and posterior (**B**; $N$ = 6 channels) thalamic ripples. SEM, standard error of the mean.
(TIF)

**S11 Fig. Thalamic ripples infrequently and weakly co-occur with cortical and hippocampal ripples during NREM.** (**A**) Cortical ripples on one channel relative to those on another ($N$ = 7,796 channel pairs). (**B**) Cortical relative to anterior thalamic ripples ($N$ = 649). (**C**) Cortical relative to posterior thalamic ripples ($N$ = 26). (**D**) Cortical relative to hippocampal ripples ($N$ = 865). (**E**) Hippocampal relative to anterior thalamic ripples ($N$ = 81). Dashed error is 98% confidence interval of the null distribution. Data are from all channel pairs from all patients. Channel pairs with significant modulations only are depicted in Fig 3. $P$-values computed using a Wilcoxon ranked-sum test to compare the modulation amplitude within −500 to 500 ms across bins and channel pairs for observed values vs. null mean values. ns = nonsignificant, ****$p < 0.0001$. NREM, non-rapid eye movement sleep. Source data are available in S11 Data.
(TIF)

**S12 Fig. Thalamic ripples rarely phase-lock with cortical and never phase-lock with hippocampal ripples during NREM.** (**A–F**) Average and SEM ΔPLV time-courses across anterior thalamo-cortical (**A**; $N$ = 1/451 significant channel pairs, post-FDR $p < 0.05$, randomization test), posterior thalamo-cortical (**B**; $N$ = 0/20), cortico-cortical (**C**; $N$ = 52/2,833 from patients 1–10 and **D**; $N$ = 7/20 from patients 11–13), hippocampo-cortical (**E**; $N$ = 0/490), and thalamo-hippocampal (**F**; $N$ = 0/47). Time-courses in color show averages across significant and in black show averages across nonsignificant channel pairs. Channel pairs were only included if they had at least 40 co-occurring ripples with a minimum overlap of 25 ms. FDR, false discovery rate; NREM, non-rapid eye movement sleep; PLV, phase-locking value; SEM, standard error of the mean.
(TIF)

**S13 Fig. Thalamo-cortical spindle phase-locking results by cortical parcel.** Mean and SEM 10–16 Hz PLVs of anterior thalamo-cortical spindles across channel pairs. Proportions of significant thalamo-cortical channel pairs are indicated for each parcel. Note the greater proportion and magnitude of significantly phase-locked thalamo-cortical channels for anterior vs. posterior cortical sites. Cortical parcels are amalgamations of the parcels in Desikan and colleagues [68], as specified in the Methods. PLV, phase-locking value; SEM, standard error of the mean.
(TIF)

**S14 Fig. Cortical spindles, cortico-cortical co-ripples, and hippocampo-cortical co-ripples are coordinated with thalamic spindles.** (**A**) Cortical spindle onsets follow thalamic spindle onsets ($N$ = 9/13 channel pairs significantly modulated, post-FDR $p < 0.05$, randomization test with shuffled controls; $N$ = 5/9 with thalamus leading cortex, two-sided binomial test comparing −500–0 ms vs. 0–500 ms, expected value = 0.5), replicated as previously found in Mak-McCully and colleagues [15]. (**B**) Cortico-cortical co-ripple centers follow thalamic spindle onsets ($N$ = 7/13 significantly modulated; $N$ = 6/7 with thalamus leading cortex). (**C**) Hippocampo-cortical co-ripple centers occur at the times of thalamic spindle onsets ($N$ = 3/9 significantly modulated; $N$ = 1/3 with thalamus leading cortex and 1/3 with cortex leading thalamus). Dashed error is 98% confidence interval of the null distribution. Plots show significant channel pairs/triplets across all patients. FDR, false discovery rate. Source data are available in S12 Data.
(TIF)

**S15 Fig. Cortical time-frequency locked to thalamic ripples.** (**A, B**) Average anterior (**A**) and posterior (**B**) thalamic-ripple locked time-frequency of cortical activity across thalamo-cortical channel pairs ($N_{aTH}$ = 649 and $N_{pTH}$ = 26). aTH, anterior thalamus; pTH, posterior thalamus.
(TIF)

**S1 Table. Patient demographics and recording characteristics.** Channels are bipolar derivations included in the study. aHC, anterior hippocampus; AV, AnteroVentral; CM, CentroMedian (of the ILTN- intralaminar nuclear complex); Fa, Fasciculus (ILTN); MD, MedioDorsal; NRT, Reticular nucleus; pHC, posterior hippocampus; PuL, Lateral Pulvinar; PuM, Medial Pulvinar; VA, Ventral Anterior; VLC, Ventral Lateral, caudal division; VLR, Ventral Lateral, rostral division; WM, white matter. Maps of cortical channels are shown in S2 Fig.
(XLSX)

**S2 Table. Pre-implantation anti-epileptic drugs by patient.** Anti-epileptic drugs were tapered off or discontinued completely within 36–48 h of admission to the epilepsy monitoring unit, where the recordings in this study were obtained. Long-acting medications, such as zonisamide, were held off prior to hospital admission preceding surgical implantation.
(XLSX)

**S3 Table. Ripple characteristics by region.** Values are counts or means and standard deviations across channels. See Fig 1 for distributions and statistics.
(XLSX)

**S1 Data. Source data.**
(XLSX)

**S2 Data. Source data.**
(XLSX)

**S3 Data. Source data.**
(XLSX)

**S4 Data. Source data.**
(XLSX)

**S5 Data. Source data.**
(XLSX)

**S6 Data. Source data.**
(XLSX)

**S7 Data. Source data.**
(XLSX)

**S8 Data. Source data.**
(XLSX)

**S9 Data. Source data.**
(XLSX)

**S10 Data. Source data.**
(XLSX)

**S11 Data. Source data.**
(XLSX)

**S12 Data. Source data.**
(XLSX)

## Acknowledgments

We thank Emília Tóth for assisting with data collection. We also thank Adam Niese, Daniel Cleary, and Jacob Garrett for their support.

## Author Contributions

**Conceptualization:** Charles W. Dickey, Sydney S. Cash, Eric Halgren.

**Data curation:** Patrick Y. Chauvel, Sandipan Pati.

**Formal analysis:** Charles W. Dickey, Ilya A. Verzhbinsky, Sophie Kajfez, Burke Q. Rosen, Christopher E. Gonzalez.

**Funding acquisition:** Eric Halgren.

**Investigation:** Charles W. Dickey, Ilya A. Verzhbinsky, Eric Halgren.

**Methodology:** Charles W. Dickey, Ilya A. Verzhbinsky, Eric Halgren.

**Resources:** Eric Halgren.

**Supervision:** Sandipan Pati, Eric Halgren.

**Visualization:** Charles W. Dickey.

**Writing – original draft:** Charles W. Dickey, Eric Halgren.

**Writing – review & editing:** Charles W. Dickey, Burke Q. Rosen, Sydney S. Cash, Sandipan Pati, Eric Halgren.

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
