## [Editor Report · Decision Letter 0]

27 Dec 2023

Dear Dr Dickey, 

Thank you for submitting your manuscript entitled "Thalamic spindles and upstates coordinate cortical and hippocampal co-ripples in humans" for consideration as a Research Article by PLOS Biology.

Your manuscript has now been evaluated by the PLOS Biology editorial staff as well as by an academic editor with relevant expertise and I am writing to let you know that we would like to send your submission out for external peer review.

Once your full submission is complete, your paper will undergo a series of checks in preparation for peer review. After your manuscript has passed the checks it will be sent out for review. To provide the metadata for your submission, please Login to Editorial Manager (https://www.editorialmanager.com/pbiology) within two working days, i.e. by Dec 29 2023 11:59PM.

Kind regards,

Christian

Christian Schnell, PhD

Senior Editor

PLOS Biology

cschnell@plos.org

---

## [Decision Letter · Decision Letter 1]

29 Jan 2024

Dear Dr Dickey,

Thank you for your patience while your manuscript "Thalamic spindles and upstates coordinate cortical and hippocampal co-ripples in humans" was peer-reviewed at PLOS Biology. It has now been evaluated by the PLOS Biology editors, an Academic Editor with relevant expertise, and by several independent reviewers. 

In light of the reviews, which you will find at the end of this email, we would like to invite you to revise the work to thoroughly address the reviewers' reports.

As you will see below, the reviewers find the study very interesting but raise a few of technical and conceptual concerns. 

Given the extent of revision needed, we cannot make a decision about publication until we have seen the revised manuscript and your response to the reviewers' comments. Your revised manuscript is likely to be sent for further evaluation by all or a subset of the reviewers.

**IMPORTANT - SUBMITTING YOUR REVISION**

*Re-submission Checklist*

*Published Peer Review*

*PLOS Data Policy*

*Blot and Gel Data Policy*

Sincerely,

Christian

Christian Schnell, PhD

Senior Editor

PLOS Biology

cschnell@plos.org

REVIEWS:

Reviewer #1: In the current manuscript, Dickey et al. report a thorough analysis of human electrophysiological recordings of thalamic, cortical, and hippocampal regions during NREM. They demonstrate ripples in the thalamus with consistent features as ripples previously reported (and reported in this study) for human cortex and hippocampus. Despite their similarity, thalamic ripples are not strongly temporally synchronized with cortical or hippocampal ripples. However, the authors show that thalamic spindles are coordinated with cortical and hippocampal spindles and ripples. 

This work is novel and important, and I find the analyses and results clearly explained in a well-written manuscript. I have only a few relatively minor analyses that are necessary to clarify or confirm some of the authors' interpretation. 

First, the authors' model (Figure 4L) indicates that thalamic spindles should precede cortical spindles/co-ripples. However, the data in Figure 4F,G appear to show that thalamic spindle power peaks shortly after the center of cortical ripples, arguing that cortical ripples are leading thalamic spindles. PLV, as far as I understand, does not provide lead/lag information, so I would like to see additional analyses directly testing the authors' model of the temporal organization of thalamic/cortical/hippocampal interactions during thalamic spindles. 

Second, the authors clearly demonstrate temporal organization of thalamic spindles with cortical ripples, but also argue for a lack of temporal organization of thalamic ripples with cortical ripples. These findings predict that thalamic spindles should not overlap with thalamic ripples. I would like to see this quantified. 

Finally, while coordination of thalamic ripples to cortical/hippocampal ripples is considerably lower than coordination of cortical-cortical ripples or cortical-hippocampal ripples, I still have questions regarding the remaining low levels of thalamic ripple/cortical ripple coordination. One hypothesis is that the cortical and thalamic networks attempt to coordinate ripples, but are unable to do so due to circuit-level limitations. However, particularly strong ripples (in either the cortex or thalamus) might be able to overcome these limitations and produce co-ripples. If the authors restrict their co-ripple analyses to only the strongest ripples, do they still fail to observe co-ripples between the cortex and thalamus (assuming they still see no volume conductance in neighboring recording locations)? 

Reviewer #2: In this study in patients, ripple events in the thalamus during NREM sleep are shown for the first time. These thalamic ripples have similar properties but are not temporally related to hippocampal and cortical ripples. The authors also replicate previous findings on the coordination of hippocampal and cortical ripples driven by thalamic spindles. The manuscript is methodologically very sound, and I commend the authors for acquiring data from thirteen epilepsy patients. However, there are some points for revision: 

Major

1. The scope of the ms is not sufficiently focused and coherent. It is the third publication of the data set and it seems like the study and analyses were not designed to fully assess the function of thalamic ripples. The authors only performed half of their analyses on these ripple events and, then, turned to replicate findings that have been shown several times in animals and humans (i.e., thalamic spindles coordinating hippocampal and cortical ripples). The authors may concentrate and extend a little more the analyses on the novel finding, i.e. thalamic ripples, for instance by:

- exploring the relationship between thalamic ripples and cortical down-to-up states and spindles. This could be interesting because the authors already describe a relationship between thalamic ripples and thalamic spindles, as well as phase-locking of thalamo-cortical spindles. 

- Since the patients were implanted two weeks prior to surgery, there might be recordings during wakefulness? It would be interesting to compare features of thalamic "wake" ripples with "sleep" ripples. Common/distinct physical features might point at similar/different functionality of these events. 

2. With the invasive methods available in humans, it is basically not possible to show that these thalamic ripple events are indeed "generated" in the thalamus. Although, using bipolar recordings between neighboring sites, in conjunction with the absence of ripples in white matter surrounding the thalamus, makes volume conductance unlikely, it still cannot be entirely excluded. Also, ripples might be travelling and entering the thalamus from other locations. Ultimately, one would need causal methods in animal models to determine where these "thalamic ripples" come from and where their precise source is. At several points in the ms it sounds as is if these ripples are actually generated in the anterior/ posterior thalamus. The discussion of these issues needs to be extended. 

Minor

1. In the time-frequency plots of Fig. 1 C ii./D there are power peaks around 500 Hz. Please explain why these peaks are present specifically in the posterior thalamus. 

2. In Fig. 2 the authors show that thalamic ripples occur during the thalamic down-to-up state. However, in the methods (p. 29) it is stated that up-states are derived from

enhanced high gamma (70-190Hz) activity. Ripples are in a similar frequency range (~120 Hz). Could this detection approach explain why more ripples were found during the up-state transition? 

3. It is proposed that thalamic ripples occur in the down-to up-state of thalamic spindles (p.8, ll.187ff). However, the results for the posterior thalamus are less clear. In Fig.2 it seems that these ripples are not locked to the up-state transition (Fig. 2H) and the results are shown for only 1 out of 6 channels. This should be states more clearly in the results/discussion section. 

Reviewer #3: In their current study, Dickey et al. investigate the role of the thalamus in coordinating co-ripple activity. This research involves 13 human subjects with medication-resistant epilepsy who underwent surgery for electrode implantation in the thalamus, hippocampus, and cortex to monitor brain activity. The study finds that ripples with a frequency of approximately 90Hz occur in both the anterior and posterior thalamus. These thalamic ripples share oscillation characteristics with those observed in the cortex and hippocampus. Notably, thalamic ripples are found to couple with local sleep spindles and precede upstates, which aligns with potential roles in memory consolidation. However, it's observed that thalamic ripples infrequently co-occur with cortical or hippocampal ripples, the latter of which often appear simultaneously across cortical sites in both hemispheres. The study also reveals that thalamo-cortical spindles and upstates significantly facilitate the co-occurrence of hippocampal-cortical ripples. The paper is nicely written and however there are some issues with the applied analysis and there is a question how novel the findings are as well as the strength of the theoretical framework they are testing.

Major comments:

1. The authors propose that co-rippling between the hippocampus and cortex might indirectly result from the thalamic connections linking these regions. Accordingly, they hypothesize that thalamic ripples may coordinate the co-occurrence of hippocampal and cortical ripples. While it is established that the thalamus has strong connections to both areas, this hypothesis requires further scrutiny. The term 'ripples,' as used in the manuscript, encompasses a variety of generation mechanisms, differing significantly between cortical and hippocampal origins. The mere presence of rippling in a third region does not necessarily imply it coordinates ripple activity between the other two regions. Additionally, the synchronization of distant brain regions is often attributed to slower oscillations, a concept supported by numerous human and rodent studies. While fast oscillations are involved in local information processing through the synchronization of neuronal activity within gamma cycles/'higher frequency oscillations', as discussed in Fries (2009). Consequently, the specific role of thalamic ripple activity in the long-distance coordination of cortical and hippocampal activity remains unclear. The authors' reliance primarily on anatomical connections to support their argument does not sufficiently address these complex dynamics.

2. Building on the previous point, the synchronization of ripple activity in both the cortex and the hippocampus may be attributed to direct coordination by slow wave activity, as discussed in Khodagoly et al. (2017, Science, surprisingly the authors do not cite this landmark paper on cortical ripples). Additionally, the potential role of the entorhinal cortex in orchestrating this activity should be considered, an aspect that was notably absent in the manuscript. While the importance of the thalamic influence on the cortex in generating spindles is undisputed, the paper falls short in exploring alternative explanations and hypotheses for these phenomena.

3. The study's findings, which suggest that spindles and slow wave oscillations coordinate the co-rippling activity in both the hippocampus and cortex, do not appear to introduce novel concepts. There seems to be a lack of clarity regarding how these current findings differentiate from those reported in Ngo et al., 2020 (published in Elife). A more detailed comparison or discussion highlighting the distinct aspects or advancements of this study, in contrast to Ngo et al.'s research, would be beneficial in understanding the unique contributions and implications of these latest results.

4. In the discussion section (lines 412-425), the authors discuss the terminology of 'ripples' and 'high frequency oscillations/high gamma oscillations.' They present an argument for generalizing the use of the term 'ripple.' While their points are noted, the discussion overlooks several important aspects related to this terminology. The generation mechanisms underlying ripples and gamma oscillations differ significantly, as do their associated cognitive correlates. Furthermore, despite some similarities between human and rodent ripples, significant differences exist, as highlighted in Liu et al. (2022, Nature Communications). In discussing the distinctions between gamma oscillations and ripples, it is crucial to reference the extensive work of John Lisman, particularly concerning gamma oscillations. The term 'high-gamma,' though deemed generic by the authors, specifically refers to gamma events that share frequency ranges with ripples. The overlapping frequency range necessitates meticulous extraction and differentiation between gamma oscillations and 'ripple' events, see Scheffer-Teixeira et al. (2012, Cerebral Cortex). Consequently, this section of the manuscript lacks accuracy and sufficient support for its claims.

5. The purpose of the paragraph in the discussion section (lines 435-444) is unclear. It raises the question: why do the authors focus on thalamic gamma oscillations? The relevance of this topic to the overall context of the paper is not evident. Moreover, this section seems disconnected from the preceding paragraph in the discussion, lacking a coherent transition or alignment with the previously established themes.

6. The authors employ Phase Locking Value (PLV) to assess the coupling between various events. However, the paper does not clearly indicate whether any surrogate analysis was conducted to rule out the possibility of chance-level occurrences among the reported events.

7. The results section (Lines 124-143) lacks the necessary quantitative and statistical data to substantiate the authors' assertions. 

8. Fig 1: why no linear axis for the frequency ranges? Makes it very hard to see the ripple frequency. For III the filtered trace should be below the raw trace since now it cannot be seen. The ripples on the raw trace are not very convincing, instead it looks like high level of high frequency noise contaminating the signal.

9. The authors should discuss why they have so few hippocampal ripples. In rodent recordings hippocampal ripples are 10times more common then cortical ripples, which is not the case in the data set.

10. Fig 2F axis title wrong?

11. Instead of only showing event triggered averages, it is also important to look at actual sequences. The authors should look at oscillation cooccurrence after detecting the different types of events and compare to shuffle controls.

12. Line 236 and onwards: how did the authors determine which cutoff is "strong" coupling?

---

## [Decision Letter · Decision Letter 2]

19 Jul 2024

Dear Dr Dickey,

Thank you for your patience while we considered your revised manuscript "Thalamic spindles and upstates coordinate cortical and hippocampal co-ripples in humans" for publication as a Research Article at PLOS Biology. This revised version of your manuscript has been evaluated by the PLOS Biology editors, the Academic Editor and the original reviewers.

Based on the reviews and on our Academic Editor's assessment of your revision, we are likely to accept this manuscript for publication, provided you satisfactorily address the remaining points raised by the reviewers. In particular, we would like to ask you to address Reviewer 3's concerns by extending the discussion of ripples, how they are defined and the role of sharp waves. We think that this would be a helpful addition to the whole "ripple" field and enhance the impact of your paper. Please also make sure to address the following data and other policy-related requests:

* We would like to suggest a different title to use the more established writing of Up states: "Thalamic spindles and Up states coordinate cortical and hippocampal co-ripples in humans" 

* Please use this spelling also in the abstract and the manuscript.

* In the abstract, please explain what Up states, spindles and co-ripples are, as our abstracts should be written for a general readership. You have enough space in the abstract to include these definitions.

* All research involving human participants must have been approved by the authors' Institutional Review Board (IRB) or an equivalent committee, and must have been conducted according to the principles expressed in the Declaration of Helsinki. Please state this (if correct) in the Methods section.

* DATA POLICY:

Regardless of the method selected, please ensure that you provide the individual numerical values that underlie the summary data displayed in the following figure panels as they are essential for readers to assess your analysis and to reproduce it: 1EFGH, 3G, 4HIJK, S4, S5, S6, S7, S8, S9, S11 and S14.

* CODE POLICY

We expect to receive your revised manuscript within two weeks. 

*Published Peer Review History*

*Press*

Sincerely,

Christian

Christian Schnell, PhD

Senior Editor

cschnell@plos.org

PLOS Biology

Reviewer remarks:

Reviewer #1: The authors have addressed all of my initial concerns. I appreciate the additional quantification and I believe the resulting manuscript is considerably improved and the authors central model is more strongly supported. 

Reviewer #2: The authors have satisfactorily addressed all my prior comments. I appreciate the clarifications on the study design and additional analyses on thalamic ripple-cortical spindle co-occurrences that are in line with their results for thalamic and cortical ripples. In the updated version of the manuscript the novelty of the findings and the scope of the study is much clearer.

Reviewer #3: The authors have only done minor changes to the manuscript and argued that including more is beyond the scope of the article. I would disagree since being the first to claim the occurrence of ripples in the thalamus does require the burden of proof. I find the added discussion on gamma vs ripples insufficient and I am not convinced by the argumentation that these events are correctly called ripples. Thus my previous comments remain unresolved. Further, the moved section did not improve readability but instead starting page 21 (tracked changes version) the discussion becomes hard to read and understand. 

I would want a detailed, critical and unbiased discussion including references beyond those generated by the last author on gamma vs ripples (how they are generated, why the sharp wave would be critical or not for the definition) and the coordination with other oscillations

---

## [Editor Report · Decision Letter 3]

20 Sep 2024

Dear Dr Dickey,

Thank you for the submission of your revised Research Article "Thalamic spindles and Up states coordinate cortical and hippocampal co-ripples in humans" for publication in PLOS Biology. On behalf of my colleagues and the Academic Editor, Guang Yang, I am pleased to say that we can in principle accept your manuscript for publication, provided you address any remaining formatting and reporting issues. These will be detailed in an email you should receive within 2-3 business days from our colleagues in the journal operations team; no action is required from you until then. Please note that we will not be able to formally accept your manuscript and schedule it for publication until you have completed any requested changes.

PRESS

Sincerely, 

Christian

Christian Schnell, PhD

Senior Editor

PLOS Biology

cschnell@plos.org